



# Developing a common, flexible and efficient framework for weakly coupled ensemble data assimilation based on C-Coupler2.0

Chao Sun[1], Li Liu[1], Ruizhe Li[1], Xinzhu Yu[1], Hao Yu[1], Biao Zhao[1,2,3], Guansuo Wang[2], Juanjuan Liu[4], Fangli Qiao[2,3], Bin Wang[1,4]

[1] Ministry of Education Key Laboratory for Earth System Modeling, Department of Earth System Science, Tsinghua University, Beijing, China

[2] First Institute of Oceanography, Ministry of Natural Resources, Qingdao, China

[3] Key Lab of Marine Science and Numerical Modeling, Ministry of Natural Resources, Qingdao, China

[4] State Key Laboratory of Numerical Modeling for Atmospheric Sciences and Geophysical Fluid Dynamics (LASG), Institute of Atmospheric Physics, Chinese Academy of Sciences, Beijing, China

*Correspondence to*: Li Liu (liuli-cess@tsinghua.edu.cn), Bin Wang (wab@tsinghua.edu.cn)

**Abstract.** Data assimilation (DA) provides better initial states of model runs by combining observational information and models. Ensemble-based DA methods that depend on the ensemble run of a model have been widely used. In response to the development of seamless prediction based on coupled models or even earth system models, coupled DA is now in the mainstream of DA development. In this paper, we focus on the technical challenges in developing a coupled ensemble DA system, which have not been satisfactorily addressed to date. We first propose a new DA framework DAFCC1 (**D**ata **A**ssimilation **F**ramework based on **C**-**C**oupler2.0, version 1) for weakly coupled ensemble DA, which enables users to conveniently integrate a DA method into a model as a procedure that can be directly called by the model. DAFCC1 automatically and efficiently handles data exchanges between the model ensemble members and the DA method, and enables the DA method to utilize more processor cores in parallel execution. Based on DAFCC1, we then develop a sample weakly coupled ensemble DA system by combining the ensemble DA system GSI/EnKF and the coupled model FIO-AOW. This sample DA system and our evaluations demonstrate the effectiveness of DAFCC1 in both developing a weakly coupled ensemble DA system and accelerating the DA system.

## 1 Introduction

Data assimilation (DA) methods, which provide better initial states of model runs by combining observational information and models, have been widely used in weather forecasting and climate prediction. The ensemble Kalman filter (EnKF; Houtekamer and Mitchell, 1998; Evensen, 2003; Lorenc, 2003a; Anderson and Collins, 2007; Whitaker, 2012) is a widely used DA method that depends on an ensemble run of members. Other DA methods that only rely on a single model run, such as three-dimensional variational analysis (3D-Var; Anderson et al., 1998; Courtier et al.1998; Gauthier et al., 1999; Lorenc, 2000) and four-dimensional variational analysis (4D-Var; Courtier et al., 1994; Kalnay, 2003; Lorenc, 2003b; Rabier et al., 2007), can be viewed as a special case of ensemble-based methods with only one member in the ensemble when we attempt to design and



develop a software framework for data assimilation. Moreover, hybrid DA methods, such as hybrid Ensemble/3D-Var (Hamill, 2000; Etherton and Bishop, 2004; Wang et al., 2008, 2013; Ma et al., 2014) and the ECMWF ensemble-based 4D-Var scheme (Fisher, 2003; Bishop and Hodyss, 2011; Bonavita et al., 2012, 2016), also depend on an ensemble run of members from the
same model.

With the rapid development of science and technology, numerical forecasting systems with only an individual component model (such as an atmospheric model) have reached a predictability limit. Coupled models have been widely used in numerical forecasting to break the bottleneck of the limited predictability, and earth system models are being used to develop seamless prediction that spans timescales from minutes to months or even decades (Palmer et al., 2008; Hoskins, 2013). Along with the
use of coupled models in numerical forecasting, common and flexible DA methods for coupled models are urgently needed (Brunet et al., 2015; Penny et al., 2017). Coupled DA technologies have already been investigated widely and DA systems have been constructed (Sugiura et al., 2008; Fujii et al., 2009, 2011; Saha et al., 2010, 2014; Sakov et al., 2012; Tardif et al., 2014, 2015; Lu et al., 2015a, b), in which ensemble based DA methods have already been applied (e.g., Zhang et al., 2005, 2007; Sluka et al., 2016).

In addition to the inherent scientific challenges of DA methods, there are technical challenges in developing a coupled ensemble DA system, which have not yet been satisfactorily addressed. These include how to conveniently (1) achieve an ensemble run of a coupled model, (2) integrate the software of a coupled model and the software of ensemble DA methods into a robust system, and (3) achieve efficient interaction between the ensemble of the coupled model and the DA methods, especially as model resolution improves. Most existing coupled ensemble DA systems such as the Data Assimilation Research
Testbed (DART; Anderson et al., 2009), the ensemble coupled data assimilation system (ECDA; Zhang et al., 2005, 2007), and Grid Point Statistical Interpolation (GSI; Shao et al., 2016) combined with EnKF (Liu et al., 2018a), employ disk files as the interfaces of data exchange between the model ensemble members and the DA methods, and iteratively switch between the run of the model ensemble and DA using software-based restart functionality that also relies on disk files. Such an implementation can guarantee software independence between the models and the DA methods, so as to achieve flexibility
and convenience in software integration; however, the I/O accesses of disk files as well as the initialization of software modules introduced by the data exchange and the restarts are time-consuming and can be a severe performance bottleneck for improved model resolution (Heinzeller et al., 2016; Craig et al., 2017). The Parallel Data Assimilation Framework (PDAF; Nerger et al., 2005; Nerger and Hiller, 2013; Nerger et al., 2019) provides a set of application programming interfaces (APIs) for transforming the DA software modules into internal procedures of the model and achieves MPI (Message Passing Interfaces)-
based data exchange between the models and DA procedures. It can therefore achieve better performance as it does not require disk files or the restart functionality. However, it still has some limitations that will be described further in Sect. 2.

To fully overcome the technical challenges, we first consider weakly coupled ensemble DA where different component models are assimilated independently from each other (Zhang et al., 2005, 2007; Fujii et al., 2009, 2011; Saha et al., 2010, 2014), and in further work will then target strongly coupled ensemble DA, which generally uses a cross-domain error
covariance matrix to account for the impact of the same observational information on different component models





cooperatively (Tardif et al., 2014, 2015; Lu et al., 2015a, b; Sluka et al., 2016). In this paper, we design and develop a common, flexible and efficient framework for weakly coupled ensemble data assimilation, based on the latest version of the Community Coupler (C-Coupler2.0; Liu et al., 2018b), that is a flexible coupler which can exchange data efficiently between different component models in a coupled system.

The remainder of this paper is organized as follows. Section 2 introduces the motivation for developing the new DA framework named DAFCC1 (**D**ata **A**ssimilation **F**ramework based on **C**-**C**oupler2.0, version 1). The overall design and implementation of DAFCC1 are described in Sect. 3 and 4, respectively. Section 5 introduces the development of a sample weakly coupled ensemble DA system based on DAFCC1. Section 6 evaluates DAFCC1. Finally, Section 7 contains a discussion and conclusions.

## 2 Motivation


The experience gained from PDAF shows that a framework with online implementation where all ensemble members of the model and all procedures of DA methods are combined into a single MPI program is essential for efficient interaction between the model and the DA software. As a coupled model and its component models often consist of general purpose software that can serve different applications other than DA systems, a DA framework should try to minimize the change to a coupled model
system when setting up its ensemble run in a single MPI program. Besides creating a working directory for each ensemble member of the coupled model, such an ensemble run requires addressing how to generate distinct local communicators from the global communicator *MPI_COMM_WORLD* of an MPI program and how to make the coupled model use a local communicator rather than the global communicator for each ensemble member. Slight modifications to the model code will be required when the original model code uses the global communicator but not a local communicator as the default. PDAF
addressed the problem of generating local communicators by imposing a precondition of process layout such that each ensemble member uses the same number of processes with successive IDs in the *MPI_COMM_WORLD*. For example, if there are 10 members in an ensemble run and each ensemble member uses 50 processes, there should be 500 processes (#0~#499) in the *MPI_COMM_WORLD*, and processes #0~#49 will be used for the first ensemble member. Such a solution can facilitate the corresponding code implementation, but requires the user to know and guarantee the process number precondition when
submitting the MPI job of an ensemble run.

In the online implementation, DA methods should be directly driven by the model, which means that DA methods should be integrated into the model as procedures that are directly called by the model driver. There are at least three issues to be addressed: 1) how to compile the code of DA methods with the model; 2) how to initialize and run the DA methods in the model execution; and 3) how to pass the corresponding information and variables between the model and DA methods.
Although model developers can fully address these questions by developing all the required code themselves, a framework should seek to minimize the required code development and to maintain independence between the DA software and the model. PDAF provides several APIs those formulate how a model initializes and runs DA methods and how the model and a DA



method exchange information and variables. However, the user is still required to perform many tasks; e.g., developing the call back functions for gathering the same model field from all ensemble members into a vector of the DA method and

distributing the data of a field in a vector to all ensemble members, according to the data structures of the model and the DA implementation. The user is recommended to make the DA implementation using the same parallel decomposition (grid domain decomposition for parallelization) as the model, and needs to develop new code when the model and DA implementation use different parallel decompositions. Moreover, efforts should be made to enable the software compilation system of the model to compile the code of the DA methods.

Most DA software consists of parallel programs that can be accelerated by using more processor cores. When running an ensemble DA algorithm for a component model in an ensemble run, all ensemble members of the component model are synchronously waiting the result of the DA algorithm. The DA algorithm therefore can be accelerated by using all the processor cores corresponding to all ensemble members of the component model. Although PDAF enables a DA algorithm to run in parallel, it only makes the processor cores of the first ensemble member available to the DA algorithm and forces the processor

cores used by other ensemble members to idle when running the DA algorithm.

## 3 Overall design of the new framework

The motivation outlined in Sect. 2 shows that further work is needed to overcome the technical challenges in developing a coupled ensemble DA system. Besides using the online implementation, our new common framework that targets weakly coupled ensemble DA as a first step should achieve the following functionalities.

1)  It can either generate the local communicators without any preconditions of process layout, or use existing local communicators that have already been generated outside the framework.

2)  It automatically handles the data exchange between the component models ensemble members and the DA algorithm to reduce the user's code development, and enables a DA algorithm to use its own compiling system for greater independence of the model and the DA algorithm.

3)  It enables a DA algorithm to use all processes of all ensemble members of the corresponding component models for acceleration.

When a DA algorithm uses the processes of all ensemble members rather than just the first ensemble member, the DA algorithm should use a parallel decomposition that differs from the model ensemble members for higher parallelism. Thus, the data exchange between the DA algorithm and a component model ensemble member will introduce a challenge of transferring

fields between the whole set of processes and a subset of processes with different parallel decompositions. Fortunately, such a challenge has already been overcome by existing couplers for earth system modelling (Craig et al., 2012; Valcke, 2012; Liu et al., 2014; Craig et al., 2017; Liu et al., 2018b). We therefore use the C-Coupler2.0 (Liu et al., 2018b), the latest version of the Community Coupler (C-Coupler), as the foundation for developing our new framework. Moreover, C-Coupler2.0 has more functionalities that the new framework can benefit from. For example, C-Coupler2.0 can handle data exchange of 3-D or even





4-D fields where the source and destination fields can have different dimension orders (e.g., vertical+horizontal at the source field, and horizontal+vertical at the destination field). It will be convenient to combine ensemble members of a coupled model into a single MPI program based on C-Coupler2.0 because each ensemble member can be registered as a component model of C-Coupler2.0, and C-Coupler2.0 can use existing local communicators of ensemble members or generate the local communicators of ensemble members without any preconditions on process layout. Most operations for achieving data exchange can be generated automatically because C-Coupler2.0 can generate coupling procedures between two sets of processes even when the two sets are overlapping.

We based the design of the architecture of our new framework on C-Coupler2.0, as shown in Fig. 1. It includes a set of new managers (i.e., DA algorithm integration manager, ensemble component manager, ensemble data exchange operation manager, online DA procedure manager, and ensemble DA configuration manager) and the new APIs corresponding to these managers. The DA algorithm integration manager enables the user to conveniently develop driving interfaces for a DA algorithm based on a set of new APIs that enables the DA algorithm to register its input and output fields and to obtain various information from the model. A DA algorithm can include a set of procedures such as observation operators and analysis modules, each of which can be called by the model separately. The framework recommends the dynamic link library (DLL) technique for the connection of a DA algorithm program to a model, so that the original configuration and compilation of a DA algorithm can generally be preserved for greater independence of the DA algorithms from the models, and for less work in integrating a DA algorithm. The ensemble component manager is responsible for generating and managing the communicator of ensemble members of a component model. The online DA procedure manager provides several APIs that enable the ensemble members of a component model to initialize, run and finalize a DA algorithm cooperatively, and automatically handles the data exchanges between the ensemble members and the DA algorithm with a set of operations. The ensemble DA configuration manager enables the user to flexibly specify the DA algorithm, DA frequency and the operations for the data exchange in a DA simulation through a configuration file.

Guided by the architecture in Fig. 1, we implemented the new framework (see Sect. 4 for detailed implementation), which enables a coupled ensemble DA system to achieve the following features that indicate that all our targets have been fully achieved.

1)  There is no restriction of process layout among ensemble members of a coupled model or among component models in each coupled model ensemble member. For example, given the three ensemble members of the coupled model that consists of four component models in Fig. 2, the first component model can take processes #0–#2, #2–#4, and #4–#6 in the first, second, and third ensemble members of the coupled model, respectively.

2)  Each component model can use different instances of DA algorithms online independently; the execution of a DA algorithm in a component model does not force the processes of other component models to be idled. For example, components 1, 2, and 4 in Fig. 2 use DA algorithms at different frequencies, while component 3 does not use DA.



3) Given a common DA algorithm, it can be used by different component models under different instances with different configurations; e.g., the fields assimilated, the observational information used, and the frequency. In Fig. 2 for example, components 2 and 4 use different instances of the same DA algorithm 2 independently.

4) An instance of a DA algorithm can either use the processes of all ensemble members of the same component model cooperatively or use the processes of each ensemble member separately. For example, each DA algorithm instance in Fig. 2 uses the processes of all ensemble members of the corresponding component model cooperatively, except procedure 1 of DA algorithm 1 that uses the processes of each ensemble member of component 1 separately.

5) Besides employing the DLL technique for integrating DA algorithm programs, scripts are allowed to conduct necessary process control, which further increases the flexibility and convenience in integrating a DA algorithm (see Sect. 4.4 for detailed implementation).

## 4 Implementation of DAFCC1

In this section, we will detail the implementation of DAFCC1 in terms of the ensemble component manager, DA algorithm integration manager, online DA procedure manager, and ensemble DA configuration manager. Moreover, we will provide an example of how to use DAFCC1 to develop a DA system.

### 4.1 Implementation of the ensemble component manager

In C-Coupler2.0, the model coupling resources, including MPI communicators, time steps, timers, model grids, parallel decompositions, coupling field instances, and coupling interfaces, are associated with each component model that is registered to C-Coupler2.0 via the API *CCPL_register_component*. When running an ensemble of a model in a single MPI run, each ensemble member should be used as a separate component model. In C-Coupler2.0, model names are the keywords to distinguish different component models. To distinguish different ensemble members of a model that generally share the same code or executable, we update the API *CCPL_register_component* to implicitly generate different names of ensemble members by appending the ID of each ensemble member to the model name (the parameter list of the API *CCPL_register_component* is unchanged). The ID of an ensemble member is given as the last argument (formatted as "CCPL_ensemble_*{ensemble numbers}_{member ID}*") of the corresponding executable when submitting an MPI run (see Fig. 3 as an example), where "*ensemble numbers*" marks the number of ensemble members and "*member ID*" marks the ensemble member ID of the current component.

Given an ensemble run of a coupled model, all ensemble members of the component models of the coupled model can be organized as one level of models (see Fig. 4a), although we recommend constructing two hierarchical levels of models with the first level corresponding to all ensemble members of the coupled model and each ensemble member including the component models at the second level (Fig. 4b), because the hierarchical organization retains the original architecture of the coupled model through a simple additional registration of the coupled model to C-Coupler2.0.





As a DA algorithm that handles ensemble fields can run on the MPI processes of all ensemble members of a component model (Fig. 2), an ensemble-set component model that covers all ensemble members of the component model is required to

use the DA algorithm (Fig. 4b). The ensemble component manager provides the capability to generate an ensemble-set component model, which does not introduce global synchronization and only involves the ensemble members of the corresponding component model.

## 4.2 Implementation of the DA algorithm integration manager

A pair of a model and a DA algorithm have essentially the relationship between a caller and a callee in a program, where the

callee generally declares a list of arguments that includes a set of input and output variables, while a caller should match the argument list of the callee when calling the callee (herein, the model is referred to as the host model of the DA algorithm). For a caller and a callee that are in the same native code, a corresponding compiler can guarantee the consistency of the argument list between them, regardless of the data structure of each argument. However, compilers cannot guarantee such consistency between a host model and a DA algorithm that is enclosed in a DLL but not in the native code of the host model.

To address the above challenge, we designed and developed a new solution for passing arguments between a host model and a DA algorithm, and tried to make such a solution as flexible as possible to improve the flexibility of DAFCC1 in serving various DA algorithms. There are three driving subroutines for initializing, running, and finalizing a DA algorithm; their subroutine names share the name of the DA algorithm as the prefix and are distinguished by different suffixes. We tried to make the explicit argument list of each driving subroutine as simple as possible (e.g., only a few integer arrays), and developed

a set of C-Coupler APIs for flexibly passing implicit arguments between the host model and the DA algorithm. Based on these APIs, the DA algorithm can obtain the required information from the host model and the grids via C-Coupler2.0 and can also declare any field instances that the DA algorithm has registered to C-Coupler2.0 as implicit input or output arguments, at the initialization stage of the DA algorithm. Figure 5 shows an example of the driving subroutines where the running and finalization driving subroutines are very simple. In the initialization driving subroutine, besides the original functionalities of

the DA algorithm such as determining parallel decompositions, allocating memory space for variables and other operations for initialization, there are additional operations for obtaining information from the host model and grids using C-Coupler2.0, registering the parallel decompositions, required grids, and field instances to C-Coupler2.0, and declaring the field instances as implicit input or output arguments.

The use of DAFCC1 requires some native code of a DA algorithm to be further updated accordingly. For example, the

original communicator of the DA algorithm needs to be replaced with the communicator of the host model that can be obtained through the corresponding C-Coupler API, and the original I/O accesses for the model data in the DA algorithm can be turned off.





## 4.3 Implementation of the online DA procedure manager

To enable different component models to use the same DA algorithm but with different configurations, a component model
can use a distinct instance of a DA algorithm with the corresponding configuration information. Corresponding to the three driving subroutines of a DA algorithm, there are three APIs (*CCPL_ensemble_procedures_inst_init*, *CCPL_ensemble_procedures_inst_run*, and *CCPL_ensemble_procedures_inst_finalize*) that enable a host model to initialize, run, and finalize the DA algorithm instance, and handle the data exchanges between the host model and the DA algorithm instance automatically. When a component model initializes, runs, or finalizes a DA algorithm instance, all ensemble members
of this component model should call the corresponding API at the same time.

### 4.3.1 API for initializing a DA algorithm instance

The API *CCPL_ensemble_procedures_inst_init* for initializing a DA algorithm instance is designed and implemented with the following steps.

1) Determine the host model of the DA algorithm instance according to the corresponding information in the configuration
file. If the DA algorithm instance is an individual algorithm that operates on the data of each ensemble member separately (e.g., Procedure 1 of DA algorithm 1 in Fig. 2), each ensemble member will be a host model. Otherwise (i.e., the DA algorithm instance is an ensemble DA algorithm that operates on the data of the ensemble set; e.g., Procedure 2 of DA algorithm 1 in Fig. 2), the host model will be the ensemble-set component model that will be generated automatically by the ensemble component manager.

2) Prepare information from the host model, such as model grids, parallel decompositions, and field instances, which the initialization driving subroutine of the DA algorithm can obtain via the corresponding APIs.

3) Initialize the corresponding DA algorithm instance according to the corresponding algorithm name and DLL name specified in the corresponding configuration file, where the corresponding DLL will be linked to the host model and the corresponding initialization driving subroutine in the DLL will be called. This implementation enables the user to
conveniently change the DA algorithms used in simulations via the configuration file without modifying the code of the model.

4) Set up data exchange operations according to the input or output fields of the DA algorithm instance declared in the initialization driving subroutine via the corresponding APIs. The data exchange is divided into two levels: the data exchange between the ensemble members and DAFCC1, and the data exchange between DAFCC1 and the DA algorithm.
The data exchange between DAFCC1 and the DA algorithm instance is simply achieved by the import/export interfaces of C-Coupler2.0, which flexibly rearrange the fields in the same component model between different parallel decompositions. If the DA algorithm instance is an ensemble algorithm, the data exchange between the ensemble members and DAFCC1 is also handled by the import/export interfaces of C-Coupler2.0, which flexibly transfer the same fields between different component models (each ensemble member and the ensemble set are different component models).





Otherwise, the data exchange between the ensemble members and DAFCC1 is simplified to a data copy. DAFCC1 will hold a separate memory space for each model field relevant to the DA algorithm, which enables a DA algorithm instance to use instantaneous model results or statistical results (i.e., mean, maximum, cumulative, and minimum) in a time window, and enables an ensemble DA algorithm instance to use aggregated results or statistical results (ensemble-mean, ensemble-anomaly, ensemble-maximum, or ensemble-minimum) from ensemble members. The sets of data exchange operations for

the input and output fields of the DA algorithm instance are generated separately.

Consistent with the functionalities in the above steps, the API *CCPL_ensemble_procedures_inst_init* includes the following arguments.

- The *ID* of the current ensemble member that calls the API, and the common full name of the ensemble members, which is used for determining the host model of the DA algorithm. When registering a component model to C-Coupler2.0, its *ID*

is allocated and its unique full name formatted as "*parent_full_name@model_name*" is generated, where "*model_name*" is the name of the component model, and "*parent_full_name*" is the full name of the parent component model (if any). Given that the names of the coupled model and the component model 1 in Fig. 4 are "*coupled*" and "*comp1*", respectively, in the one-level model hierarchy in Fig. 4a, the full names of ensemble members of the component model 1 are "*comp1_1*" to "*comp1_N*" and the common full name is "*comp1_**" where "*" is a wildcard, while in the two-level model hierarchy

in Fig. 4b the full names of ensemble members of the component model 1 are "*coupled_1@comp1*" to "*coupled_N@comp1*" and the common full name is "*coupled_*@comp1*".

- The name of the DA algorithm instance, which is the keyword of the DA algorithm instance and also specifies the corresponding configuration information. Different DA algorithm instances can correspond to different DA algorithms or the same DA algorithm. For example, the component models 2 and 4 use different instances of the same DA algorithm in

Fig. 2.

- A list of model grids and parallel decompositions, which are optional arguments that enable the DA algorithm instance to obtain grid data and use the same parallel decompositions as the host model.

- A list of field instances, which specify the model fields that can be used for assimilation. This list should cover all input or output fields of the DA algorithm.

- An optional integer array of control variables that can be obtained by the DA algorithm instance via the corresponding APIs.

- An annotation, which is a string giving a hint for locating the model code of the API call corresponding to an error or warning, is recommended but not mandatory, and should be provided by the user.

### 4.3.2 API for running a DA algorithm instance

The API *CCPL_ensemble_procedures_inst_run* is responsible for running a DA algorithm instance with the following steps.

1) Executing the data exchange operations for the input fields of the DA algorithm instance. This step automatically transfers the input fields from each ensemble member of the corresponding component model to DAFCC1 and then from DAFCC1





to the DA algorithm instance, where the statistical processing regarding the time window or the ensemble is done at the same time.

2)   Executing the DA algorithm instance through calling the running driving subroutine of the DA algorithm.

      3)   Executing the data exchange operations for the output fields of the DA algorithm instance. This step automatically transfers the output fields from the DA algorithm instance to DAFCC1 and then from DAFCC1 to each ensemble member of the corresponding component model.

Each DA algorithm instance has a timer specified via the configuration information, which determines when the DA

algorithm instance is run. The *CCPL_ensemble_procedures_inst_run* can be called for the DA algorithm instance at each time step, while the above three steps will be executed only when the corresponding timer is on. To store the input data such as the observational information, a DA algorithm instance can either share the working directory of its host model or use its own working directory specified via the configuration information. The API *CCPL_ensemble_procedures_inst_run* will change and then recover the current directory for calling the running driving subroutine of the DA algorithm, if necessary.

**4.3.3 API for finalizing a DA algorithm instance**

The API *CCPL_ensemble_procedures_inst_finalize* is responsible for finalizing a DA algorithm instance through calling the finalization driving subroutine of the DA algorithm.

**4.4 Implementation of the ensemble DA configuration manager**

The configuration information of all DA algorithm instances used in a coupled DA simulation is contained in an XML

configuration file (e.g., Fig. 6), and each DA algorithm instance has a distinct XML node (e.g., the XML node "da_instance" in Fig. 6, where the attribute "name" is the name of the DA algorithm instance and also the keyword to match the name of the DA algorithm instance in API "*CCPL_ensemble_procedures_inst_init*"), which enables the user to specify the following configurations.

  1)   The DA algorithm specified in the XML node "external_procedures" in Fig. 6, where the attribute "dll_name" specifies

310       the dynamic link library, and the attribute "procedures_name" specifies the name of the DA algorithm, which will be used to choose the driving subroutines. When the user seeks to change the DA algorithm used by a component model, it is only necessary to modify the XML node "external_procedures" in most cases.

  2)   The periodic timer specified in the XML node "periodic_timer" in Fig. 6. Besides the attribute "period_unit" and "period_count" for specifying the period of the timer, the user can specify a lag via the attribute "local_lag_count". For

315       example, given a periodic timer <"period_unit"="hours", "period_count"=6, "local_lag_count"=3>, its period is 6 hours, and it will not be on at the $0^{th}$, $6^{th}$, and $12^{th}$ hours, but instead on at the $3^{rd}$, $9^{th}$, and $15^{th}$ hours due to the "local_lag_count" of 3.

  3)   Statistical processing of input fields specified in the XML node "field_instances" in Fig. 6, where the attribute "time_processing" specifies the statistical processing in each time window determined by the periodic timer and the





attribute "ensemble_operation" specifies the statistical processing among ensemble members. For an individual DA algorithm, the attribute "ensemble_operation" should be set to "*none*". Besides the default specification of statistical processing shared by all fields, a field can have its own statistical processing specified in a sub node of the XML node "field_instances".

4)    The working directory and the scripts for pre- and post-assimilation analysis (e.g., for processing the data files of

observational information) optionally specified in the XML node "processing_control" in Fig. 6. When the working directory is not specified, the DA algorithm instance will use the working directory of its host model. The script specified in the sub XML node "pre_instance_script" will be called by the root process of the host model before the API *CCPL_ensemble_procedures_inst_run* calls the DA algorithm, and the script specified in the sub XML node "post_instance_script" will be called by the root process of the host model after the DA algorithm run finishes.

## 5 A sample weakly coupled ensemble DA system based on DAFCC1


To provide further information on how to use DAFCC1 and for validating and evaluating DAFCC1, we developed a sample weakly coupled ensemble DA system by combining the ensemble DA system GSI/EnKF (Shao et al., 2016; Liu et al., 2018b) and a regional Atmosphere-Ocean-Wave coupled model which is referred as FIO-AOW (Zhao et al., 2017; Wang et al., 2018). GSI/EnKF mainly focuses on regional numerical weather prediction (NWP) applications coupled with the Weather Research

and Forecasting (WRF) model (Wang et al., 2014), while FIO-AOW consists of WRF, the Princeton Ocean Model (POM; Blumberg and Mellor 1987; Wang et al., 2010), the MArine Science and NUmerical Modeling wave model (MASNUM; Yang et al., 2005; Qiao et al, 2016), and all the above three model components are coupled together by using C-Coupler (Liu et al., 2014, 2018b). There are two main steps in developing the sample system.

1)    We developed an ensemble DA sub-system of WRF by adapting GSI/EnKF to DAFCC1. This sub-system helps validate

DAFCC1 and evaluate the improvement in performance obtained by DAFCC1 (Sect. 6).

2)    We merged the above sub-system and FIO-AOW to produce the sample DA system, which demonstrates the effectiveness of DAFCC1 in developing a weakly coupled ensemble DA system.

### 5.1 An ensemble DA sub-system of WRF

### 5.1.1 Brief introduction to GSI/EnKF

GSI/EnKF combines a variational DA sub-system (GSI; Shao et al., 2016) and an ensemble DA sub-system (EnKF; Liu et al., 2018a). It provides two options for calculating analysis increments for ensemble DA; i.e., a serial Ensemble Square Root Filter (EnSRF) algorithm (Whitaker et al., 2012) and a Local Ensemble Kalman Filter (LETKF) algorithm (Hunt et al., 2007). It can be used as a pure ensemble DA system without using variational DA, where GSI is used as the observation operator that calculates the difference between model variables and observations on the observation space and EnKF calculates analysis





increments and updates model variables. In this paper, we focus on the pure ensemble DA system for developing the sample DA system.

Figure 7a shows the flowchart for running the pure ensemble DA system in a DA window. It consists of the following steps that are driven by scripts, while the data exchanges between steps are achieved via data files.

1) Ensemble model forecast. An ensemble run of WRF is initiated or restarted from a set of input data files, and then is

stopped after producing a set of output files (called model background files hereafter) for DA and for restarting the ensemble run in the next DA window.

2) Calculating the ensemble mean of model DA variables. A separate executable is initiated for calculating the ensemble mean of each DA variable based on the model background files, and then outputs the ensemble mean to a new background file.

3) Observation operator for the ensemble mean. GSI is initiated as the observation operator for the ensemble mean. It takes the ensemble mean file, files of various observational data (e.g., conventional data, satellite radiance observations, GPS radio occultations, and radar data) and multiple fixed files (e.g., statistic files, configuration files, bias correction files, and CRTM coefficient files) as input, and produces an observation prior (observation innovation) file for the ensemble mean and files containing observational intermediate information (e.g., bias correction and thinning).

4) Observation operator for each ensemble member. GSI is initiated as the observation operator for each ensemble member. It takes the background file of the corresponding ensemble member, the fixed files and the observational intermediate information files as input, and produces an observation prior file for the corresponding ensemble member.

5) EnKF for calculating analysis increments. EnKF is initiated for calculating analysis increments of the whole ensemble. It takes the model background files, the observation prior files and the fixed files as input, and finally updates model

background files with the analysis increments. The updated model background files are used for restarting the ensemble model forecast in the next DA window.

### 5.1.2 Adapting GSI/EnKF to DAFCC1

To adapt GSI/EnKF to DAFCC1, we used the WRF model for the first main step of the ensemble forecast as the host model that drives the DA algorithm instances corresponding to the remaining main steps. As shown in Fig. 8, there are three DA

instances corresponding to the last three main steps, without the DA algorithm instance corresponding to the second main step. This is because the online DA procedure manager of DAFCC1 enables a DA algorithm instance to automatically obtain the ensemble mean of model DA variables (Sect. 4.3). Although both the third and fourth main steps correspond to the same GSI, they are transformed into two different DA algorithm instances, because the third is an ensemble algorithm (i.e., it operates on the data of the ensemble set) and the fourth is an individual algorithm (i.e., it operates on the data of each ensemble member).

Moreover, we compiled the same GSI code into two separate DLLs, each of which corresponds to one of these two instances, to enable these two instances to use different memory space.





For each DA algorithm instance, three driving subroutines and the corresponding configuration were developed. In fact, the two instances corresponding to GSI share the same driving subroutines but use different configurations (especially regarding the specification of "ensemble_operation"). To enable the GSI code and EnKF code to be used as DLL, we made the following

slight modifications to the code.

1)    We turned off the MPI initialization and finalized and replaced the original MPI communicator with the MPI communicator of the host model that can be obtained via DAFCC1.

2)    We obtained the required model information and the declared input/output fields via DAFCC1, and turned off the corresponding I/O accesses.

To drive the DA algorithm instances, the WRF code was updated with the new subroutines for initializing, running, and finalizing all DA algorithm instances. Moreover, the functionality of outputting model background files can be turned off, because the data exchanges between WRF and the DA algorithm instances are automatically handled by DAFCC1 and the WRF ensemble can be run continuously throughout DA windows without stopping and restarting. As a result, DAFCC1 saves sets of data files and the corresponding I/O access operations, while only the observation files, fixed files, and the files for the

data exchanges among the DA algorithm instances are reserved (compare Fig. 7b and Fig. 7a).

## 5.2 Sample ensemble DA system of FIO-AOW

FIO-AOW, which previously used C-Coupler1 (Liu et al., 2014) for model coupling, has already been upgraded to C-Coupler2.0. As GSI/EnKF and FIO-AOW share WRF, the development of the sample ensemble DA system of FIO-AOW can significantly benefit from the DA system of WRF, and it only took the following steps to construct the sample ensemble DA

system.

1)    Using the ensemble component manager, set up the two hierarchical levels of models shown in Fig. 9; i.e., the first level corresponds to all ensemble members of FIO-AOW while each member includes its three component models at the second level.

2)    Merge the model code modifications, the DA algorithm instances, and configurations in the DA system of WRF into the

405        sample ensemble DA system FIO-AOW.

As well as being described by the flowchart involving the WRF and the DA algorithm instances in Fig. 7b, the sample ensemble DA system of FIO-AOW follows the process layout in Fig. 10, which is essentially a real case of the process layout in Fig. 2.

## 6 Validation and evaluation of DAFCC1

In this section, we evaluate the effectiveness of DAFCC1 in developing a weakly coupled ensemble DA system based on the sample ensemble DA system (referred to as the full-sample-DA-system hereafter) described in Sect. 5, and will also validate





DAFCC1 and evaluate the impact of DAFCC1 in accelerating DA based on the sub-system with WRF and GSI/EnKF (WRF-GSI/EnKF hereafter).

## 6.1 Experimental setup

The sample ensemble DA system used in this validation and evaluation consists of WRF Version 4.0 (Wang et al., 2014), GSI version 3.6 and EnKF version 1.2, and the corresponding versions of POM and MASNUM used in FIO-AOW (Zhao et al., 2017; Wang et al., 2018). In EnKF version 1.2 the default settings are used; i.e., the EnSRF algorithm is used to calculate analysis increments for ensemble DA, the inflation factor is 0.9 without smoothing, and the covariance is localized by distance correlation function with horizontal localization radius of 400 km and vertical localization scale coefficient of 0.4. The sample

ensemble DA system is run on a supercomputer of the Beijing Super Cloud Computing Center (BSCC). Each computing node on the supercomputer includes two Intel Xeon E5-2678 v3 CPUs (Intel(R) Xeon(R) CPU), with 24 processor cores in total, and all computing nodes were connected with an InfiniBand network. The codes were compiled by an Intel Fortran and C++ compiler at the optimization level O2, using an Intel MPI library. A maximum 3200 cores are used for running the sample ensemble DA system.

The WRF-GSI/EnKF integrates over an approximate geographical area generated from a Lambertian projection of the area 0°–50°N, 99°–160°E with center point at 35°N, 115°E. Initial fields and lateral boundary conditions (at 6 hour intervals) for the ensemble run of WRF are taken from the NCEP Global Ensemble Forecast System (GEFS) (at 1° × 1° resolution) (https://www.ncdc.noaa.gov/data-access/model-data/model-datasets/global-ensemble-forecast-system-gefs). To configure WRF, an existing physics suite 'CONUS' (https://www2.mmm.ucar.edu/wrf/users/ncar_convection_suite.php) and 32 vertical

sigma layers with the model top at 50 hPa are used. One-day integration on June 1st, 2016 is used for running the WRF-GSI/EnKF. NCEP global GDAS Binary Universal Form for the Representation of meteorological data (BUFR; https://www.emc.ncep.noaa.gov/mmb/data_processing/NCEP_BUFR_File_Structure.htm) and Prepared BUFR (https://www.emc.ncep.noaa.gov/mmb/data_processing/prepbufr.doc/document.htm), including conventional observation data and satellite radiation data, are assimilated every 6 hours (i.e., at 0000, 0600, 1200, and 1800 UTC). The air temperature

(T), specific humidity (QVAPOR), longitude and latitude wind (UV), and column disturbance dry air quality (MU) are the variables used for DA. The WRF-GSI/EnKF experiments are classified into four sets, where variations of horizontal resolution (and the corresponding time step), number of ensemble members and process number (each process runs on a distinct processor core) are considered (Tables 1 and 2).

All component models of the full-sample-DA-system integrate over the same geographical area (0°–50°N, 99°–150°E) with

the same horizontal resolution of 0.5° × 0.5° but different time steps (100 s for WRF and 300 s for POM and MASNUM, coupled by C-Coupler2.0 at 300 s intervals). More details of the model configurations can be found in Zhao et al. (2017). The configuration of initial fields, lateral boundary conditions, and observations of WRF for the ensemble run of the full-sample-DA-system are the same as for WRF-GSI/EnKF. The full-sample-DA-system integrates over 3 days (June 1st to 3rd, 2016),





while the first model day is considered as spin-up, and DA is performed every 6 hours in the last two model days with T, UV
and MU as DA variables.

## 6.2 Validation of DAFCC1

To validate DAFCC1, we compare the outputs of the two versions of WRF-GSI/EnKF: the original WRF-GSI/EnKF (hereafter offline WRF-GSI/EnKF; https://dtcenter.org/community-code/gridpoint-statistical-interpolation-gsi/community-gsi-version-3-6-enkf-version-1-2) and the new version of WRF-GSI/EnKF with DAFCC1 (hereafter online WRF-GSI/EnKF) introduced

in Sect. 5.1. As DAFCC1 improves only the data exchanges between a model and the DA algorithms, the simulation results of an existing DA system should not change when it is adapted to use DAFCC1. We therefore employ a bitwise identical standard for validating DAFCC1, which means the online WRF-GSI/EnKF achieves exactly the same result as the offline WRF-GSI/EnKF. DAFCC1 passes the validation test with all experimental setups in Table 2, where the binary data files output by WRF at the end of the 1-day integration are used for the comparison.

## 455 6.3 Impact in accelerating DA

WRF-GSI/EnKF is further used to evaluate the impact of DAFCC1 in accelerating DA, by comparing the execution time of the offline and online WRF-GSI/EnKF under each experimental setup in Table 2. Considering that all ensemble members of the online WRF-GSI/EnKF are integrated simultaneously, we run all ensemble members of the offline WRF-GSI/EnKF concurrently through a slight modification to the corresponding script, in order to make a fair comparison.

The impact of varying the number of ensemble members is evaluated based on Set 1 in Table 2. DAFCC1 obviously accelerates WRF-GSI/EnKF, and can achieve higher performance speedup with more ensemble members (Fig. 11a). This is because DAFCC1 significantly accelerates the DA for both GSI and EnKF (Fig. 11b-d). Similarly, DAFCC1 significantly accelerates the DA as well as WRF-GSI/EnKF under different process numbers (Fig. 12, corresponding to Set 2 in Table 2) and resolution (Fig. 13, corresponding to Set 3 in Table 2). Considering that more processor cores are generally required to

accelerate the model run under higher resolution, we also make an evaluation based on Set 4 in Table 2, where concurrent changes in resolution and process number are made to achieve similar numbers of grid points per process throughout the experimental setups. This evaluation also demonstrates the effectiveness of DAFCC1 in accelerating the DA as well as WRF-GSI/EnKF (Fig. 14).

The performance speedups observed from Figs. 11–14 result mainly from the significant decrease in I/O accesses. Although

the online WRF-GSI/EnKF still has to access the observation prior files (Sect. 5.1.1 and Fig. 7b), most I/O accesses correspond to the model ensemble background files and model ensemble analysis files, and these I/O accesses have been eliminated by DAFCC1 (Table 3). Moreover, more I/O accesses can be saved under higher resolution or more ensemble members.





### 6.4 Effectiveness in developing a weakly coupled ensemble DA system

We have successfully run the full-sample-DA-system with ten ensemble members, which enables us to investigate the model
variables before and after DA. We find that changes to the model variables resulting from DA can be observed; e.g., the bias
regarding T is slightly decreased and the bias regarding UV is more obviously decreased after using DA, as shown in Fig. 15.

### 7 Conclusions and discussion

In this paper, we propose a new common, flexible and efficient framework for weakly coupled ensemble data assimilation
based on C-Coupler2.0, DAFCC1. It provides simple APIs and a configuration file format to enable users to conveniently
integrate a DA method into a model as a procedure that can be directly called by the model, while still guaranteeing software
independence between the model and the DA method. The sample weakly coupled ensemble DA system in Sect. 5 and the
evaluations in Sect. 6 demonstrate the effectiveness of DAFCC1 in both developing a weakly coupled ensemble DA system
and accelerating the DA system.

DAFCC1 is able to automatically handle data exchanges between a model ensemble and a DA algorithm because its design
and implementation significantly benefit from C-Coupler2.0, which already has the functionalities of automatic coupling
generation and automatic data exchanges between different component models or within the same component model. DAFCC1
will therefore be an important functionality of the next generation of C-Coupler (C-Coupler3) which is planned to be released
no later than 2022. We have considered software extendibility when designing and implementing DAFCC1, which will enable
us to conveniently achieve upgrades either for strongly coupled ensemble DA systems or for more types of data exchange
operations in the future. As shown in Fig. 7, the I/O accesses to the observation prior files for the data exchanges between DA
algorithms are still retained after using DAFCC1. Although they are not currently a performance bottleneck (Table 3), we will
investigate how to avoid these types of I/O accesses when further upgrading DAFCC1.

Regarding the evaluations in Sect. 6, we can only use at most 3200 processor cores, which limits the maximum number of
cores per ensemble member. Consequently, we use relatively coarse resolutions of WRF and FIO-AOW. However, the results
in Fig. 14 from the experiment Set 4 in Table 2 indicate that DAFCC1 will also obviously accelerate the DA system when
using a finer resolution and more processor cores, because it will also significantly decrease I/O accesses. DAFCC1 can tackle
the technical challenges in developing or accelerating a DA system, but cannot contribute to improvements in simulation
results that generally depend on scientific settings which must be determined in the research environment (e.g., the DA
algorithm configuration, the inflation factor, localization settings, initial states of the model ensemble run). Consequently, we
did not examine the improvements in simulation results resulting from the full-sample-DA-system based on various variables
in Sect. 6.4, but only made a simple comparison of simulation results demonstrating that the full-sample-DA-system can
successfully run and produce simulation results.



*Code availability.* The source code of DAFCC1 can be viewed via https://doi.org/10.5281/zenodo.3739729 (please contact us for authorization before using DAFCC1 for developing a system). The original source code and scripts corresponding to WRF

and GSI/EnKF can be download from https://www2.mmm.ucar.edu/wrf/users/downloads.html and https://dtcenter.org/com-GSI/users/downloads/index.php respectively. For the source code of FIO-AOW, please contact the authors of (Zhao et al., 2017; Wang et al., 2018). The additional codes, configurations, scripts and guidelines for developing and running the sample weakly coupled ensemble DA system can also be download from https://doi.org/10.5281/zenodo.3774710.

*Author contributions.* CS was responsible for code development, software testing and experimental evaluation of DAFCC1

with the sample DA system, contributed to the motivation and design of DAFCC1 and co-led paper writing. LL initiated this research, was responsible for the motivation and design of DAFCC1, co-supervised CS, and co-led paper writing. RL, XY and HY contributed to code development and software testing. BZ, GW, JL and FQ contributed to the development of the sample DA system. BW contributed to scientific requirements and the motivation, and co-supervised CS. All authors contributed to improvement of ideas and paper writing.

*Competing interests.* The authors declare that they have no conflict of interest.

*Acknowledgements.* This work was jointly supported in part by the National Key Research Project of China (grant no. 2017YFC1501903), the National Natural Science Foundation of China(grant no. 41875127) and the National Key Research Project of China (grant no. 2016YFA0602204).

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






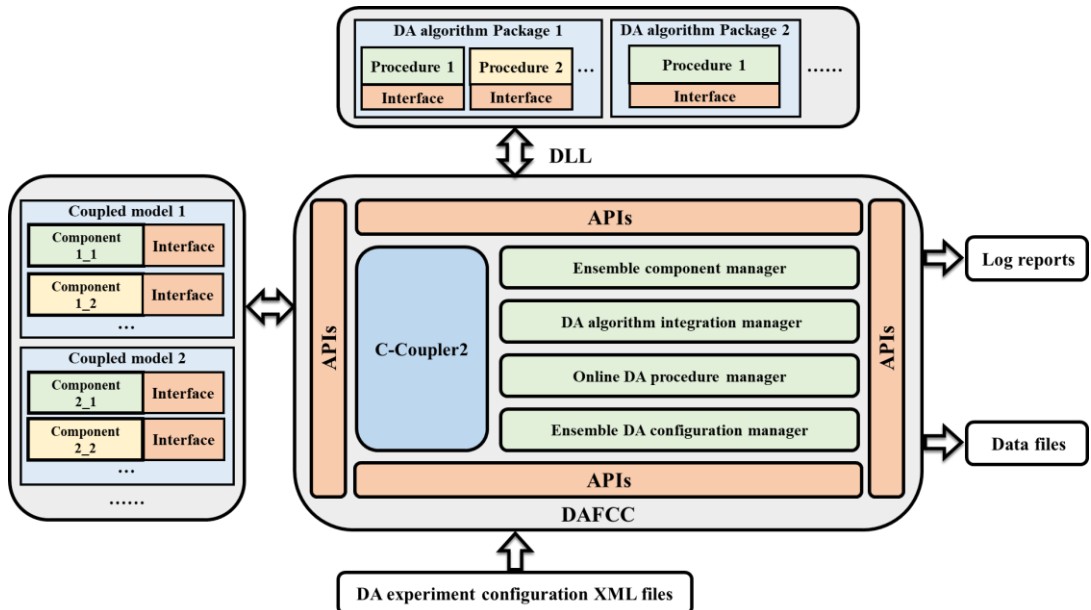

**Figure 1. Architecture of DAFCC1.**





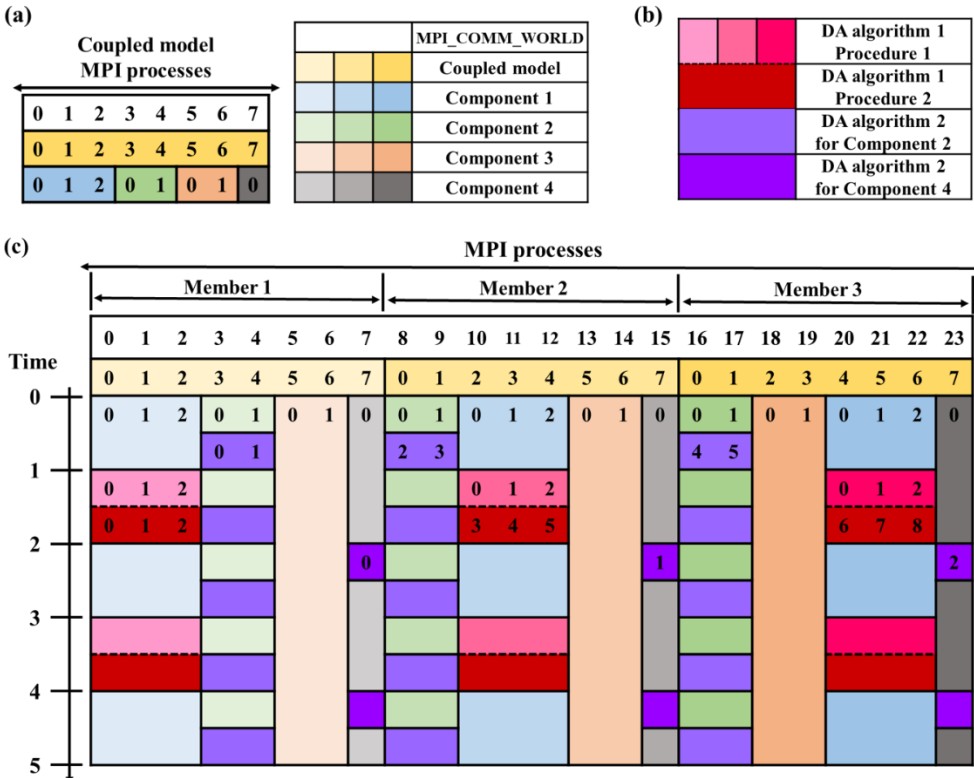

**Figure 2. Example of running a DAFCC1-based weakly coupled ensemble DA system with three ensemble members. (a) Each ensemble member of the coupled model (yellow series) uses 8 MPI processes, where component 1 (blue series) uses three MPI processes, component 2 (green series) uses two MPI processes, component 3 (orange series) uses two MPI processes, and component 4 (grey series) uses one MPI process. (b) DA algorithm 1 and two instances of DA algorithm 2 (purple series) are used in this DA system, where DA algorithm 1 includes procedure 1 (pink series) and procedure 2 (red). (c) Execution of the DA system: the process layout of ensemble members of component models, the process layout of DA algorithms, and the alternative execution of a DA algorithm and the corresponding component model. Each number in the colored box in (a) and (c) indicates the process ID in the corresponding local communicator of a member of the coupled model, a member of a component model, or all members of a component model.**

```
mpirun –np N1_1 Comp1 namelist CCPL_ensemble_3_1 : –np N1_2 Comp2 namelist CCPL_ensemble_3_1 : –np N2_2
Comp2 namelist CCPL_ensemble_3_2 : –np N2_1 Comp1 namelist CCPL_ensemble_3_2 : –np N3_1 Comp1 namelist
CCPL_ensemble_3_3 : –np N3_2 Comp2 namelist CCPL_ensemble_3_3
```

**Figure 3. Example of the command for submitting an MPI run of three ensemble members of a coupled model that consists of Comp1 and Comp2. Comp1 can be before Comp2 at the second ensemble member, and the process numbers N1_1, N2_1, and N3_1 of Comp1 at different ensemble members can be different, because C-Coupler2.0 does not introduce any preconditions on the process layout of ensemble members of models.**





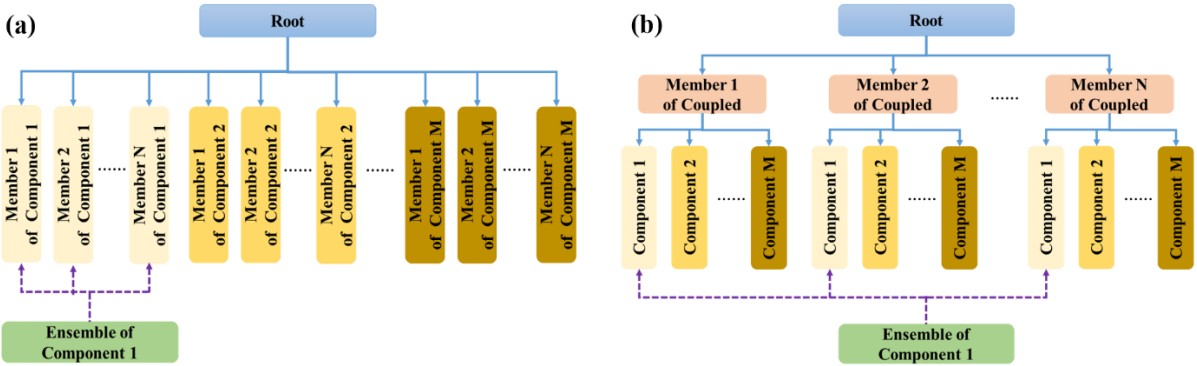


**Figure 4. Two examples of the organization of N ensemble members of a coupled model consisting of M component models. (a) Single-level organizational architecture of all ensemble members of the component models in the coupled model. (b) Two-hierarchical-levels organizational architecture. All ensemble members of the coupled model are organized as the first level with all component models from each ensemble member of the coupled model at the second level. An ensemble that covers all ensemble members of component model 1 is generated as an example for using the DA algorithm in ensemble component manager.**


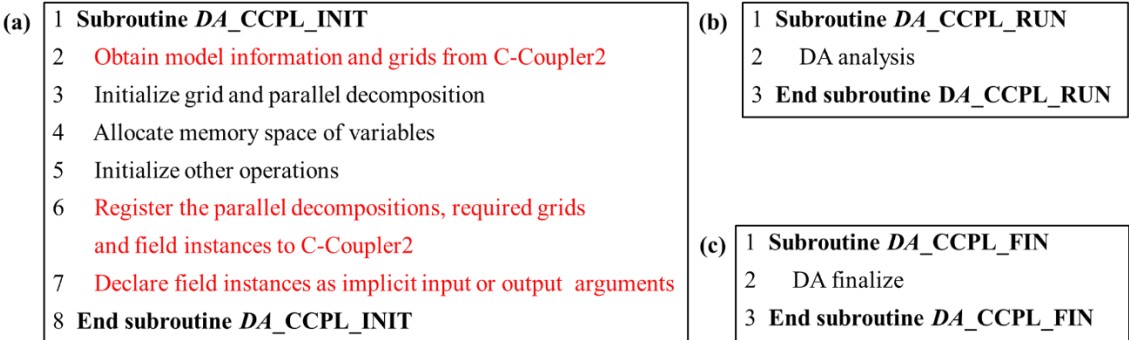

**Figure 5. Example of the driving subroutines in a DA algorithm. (a) Initialization driving subroutine. (b) Running driving subroutine. (c) Finalization driving subroutine. The name of the DA algorithm "DA" is used as the prefix of the three driving subroutines; different suffixes are used for distinction. Black font indicates original functionalities of the DA algorithm, while red font indicates additional operations to perform online data exchanges between the model and DA algorithm.**






```
<root>
<da_instance name="DA_algorithm1_procedure1" status="on">
<external_procedures status="on" procedures_name="algorithm1_procedure1" dll_name="lib_da1_p1.so"/>
<periodic_timer status="on" period_unit="seconds" period_count="21600" local_lag_count="0"/>
<field_instances status="on" time_processing="inst" ensemble_operation="none"/>
<processing_control status="on" >
<working_directory status="off" path=""/>
<config_scripts status="on">
<pre_instance_script status="on" name="da1_p1_online_run.sh"/>
<post_instance_script status="off" name=""/>
</config_scripts>
</processing_control>
</da_instance>
<da_instance name="DA_algorithm1_procedure2" status="on">
<external_procedures status="on" procedures_name="algorithm1_procedure2" dll_name="lib_da1_p2.so"/>
<periodic_timer status="on" period_unit="seconds" period_count="21600" local_lag_count="0"/>
<field_instances status="on" time_processing="inst" ensemble_operation="aver">
</field_instances>
<processing_control status="on" >
<working_directory status="on" path="./experiment/da1"/>
<config_scripts status="on">
<pre_instance_script status="off" name=""/>
<post_instance_script status="on" name="da1_p2_online_run.sh"/>
</config_scripts>
</processing_control>
</da_instance>
<da_instance name="DA_algorithm2" status="on">
<external_procedures status="on" procedures_name="algorithm2" dll_name="lib_da2.so"/>
<periodic_timer status="on" period_unit="seconds" period_count="21600" local_lag_count="0"/>
<field_instances status="on" time_processing="inst" ensemble_operation="gather"/>
<processing_control status="on" >
<working_directory status="on" path="./experiment/da2"/>
<config_scripts status="on">
<pre_instance_script status="on" name="./scripts/da2_pre_online_run.sh"/>
<post_instance_script status="on" name="./scripts/da2_post_online_run.sh"/>
</config_scripts>
</processing_control>
</da_instance>
</root>
```

**Figure 6. Example of the XML configuration for a DA experiment.**



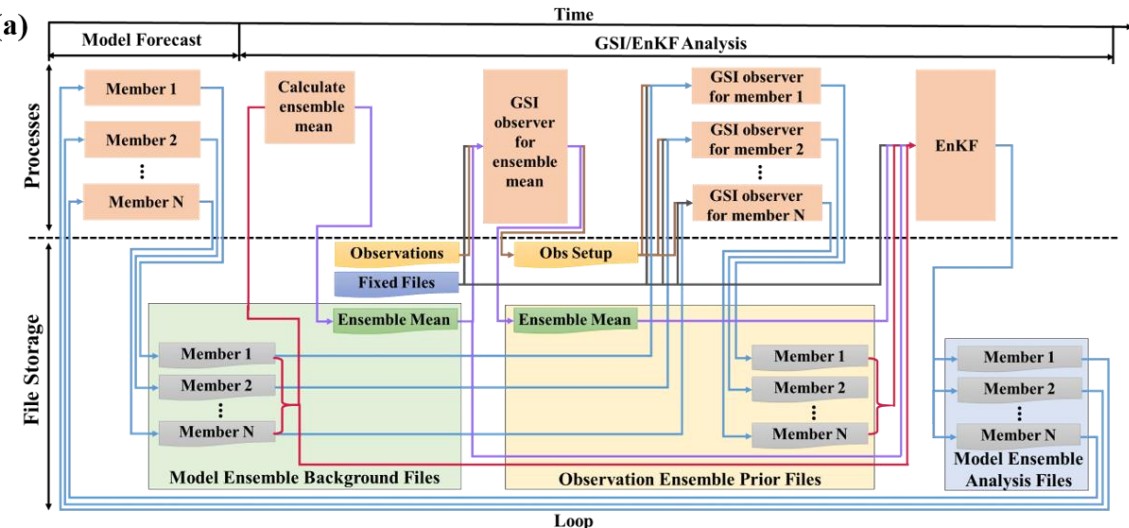

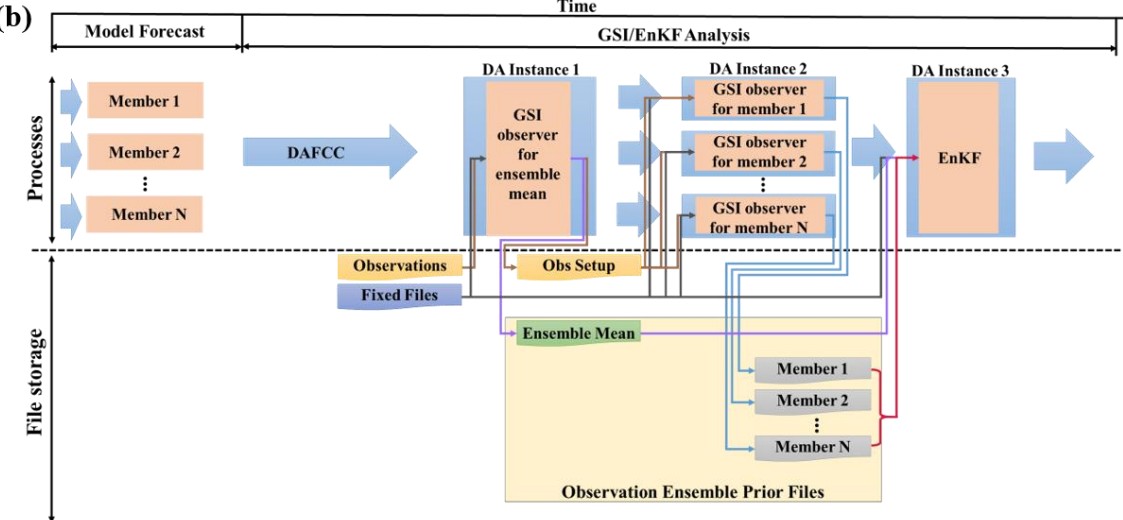

**Figure 7. Running processes and data scheduling for (a) original GSI/EnKF used as a pure ensemble DA system, and (b) modified GSI/EnKF based on DAFCC1. Orange rectangles in the Processes panel indicate different running processes, while thick blue arrows mark data scheduling based on DAFCC1. Rectangles of various colors with a curved lower edge in the File Storage panel indicate different files, while arrows of different colors indicate the scheduling of corresponding files.**





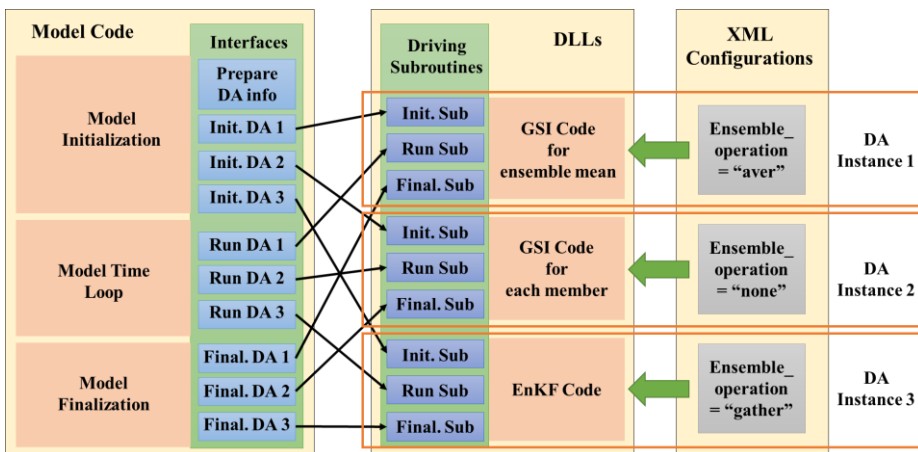

**Figure 8. Modifications of model code and the invoking of relationships to the DA algorithm in the sample ensemble DA system.**

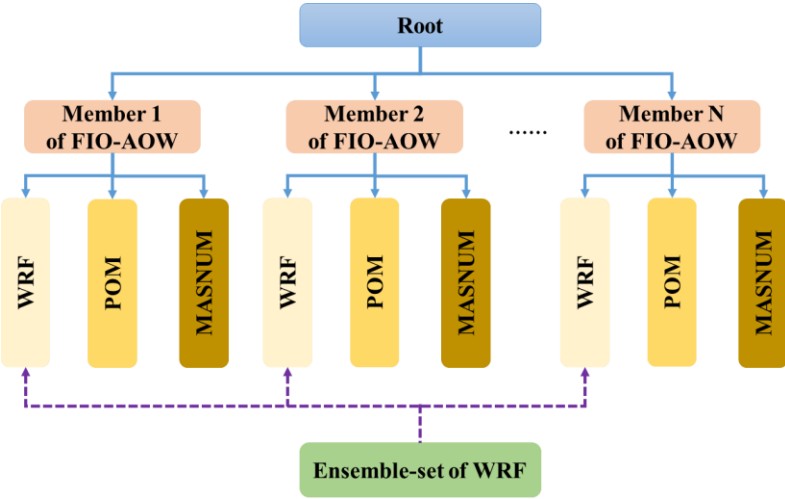


**Figure 9. Two-hierarchical-level organizational architecture for N ensemble members of FIO-AOW consisting of WRF, POM, and MASNUM. All ensemble members of FIO-AOW are organized as the first level with all component models in each ensemble member at the second level. An ensemble-set that covers all ensemble members of component model WRF is generated by the ensemble component manager.**




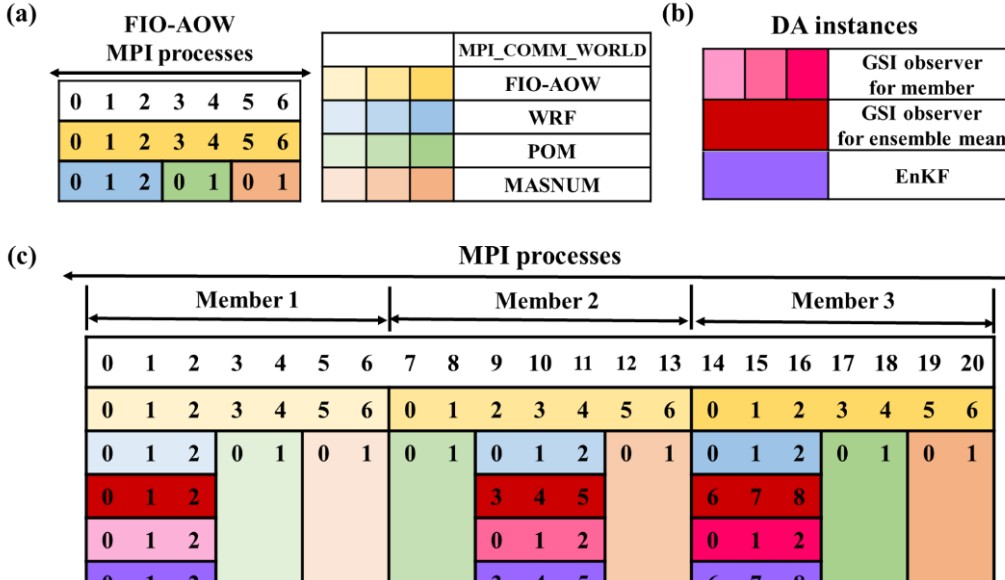

Figure 10. Example of the process layout of the sample ensemble DA system FIO-AOW. (a) Each ensemble member of FIO-AOW (yellow series) uses 7 MPI processes, where WRF (blue series) uses 3 MPI processes, POM (green series) uses 2 MPI processes, and MASNUM (orange series) uses 2 MPI processes. (b) Two DA algorithm instances of GSI are adopted for each member (pink series) and ensemble mean (red) respectively following another DA algorithm instance of EnKF in this DA system. (c) Process layout of the DA system: the process layout of ensemble members of component models and the process layout of DA algorithms. Each number in the colored boxes in (a) and (c) indicates the process ID in the corresponding local communicator of a member of the coupled model, a member of a component model, or all members of a component model.




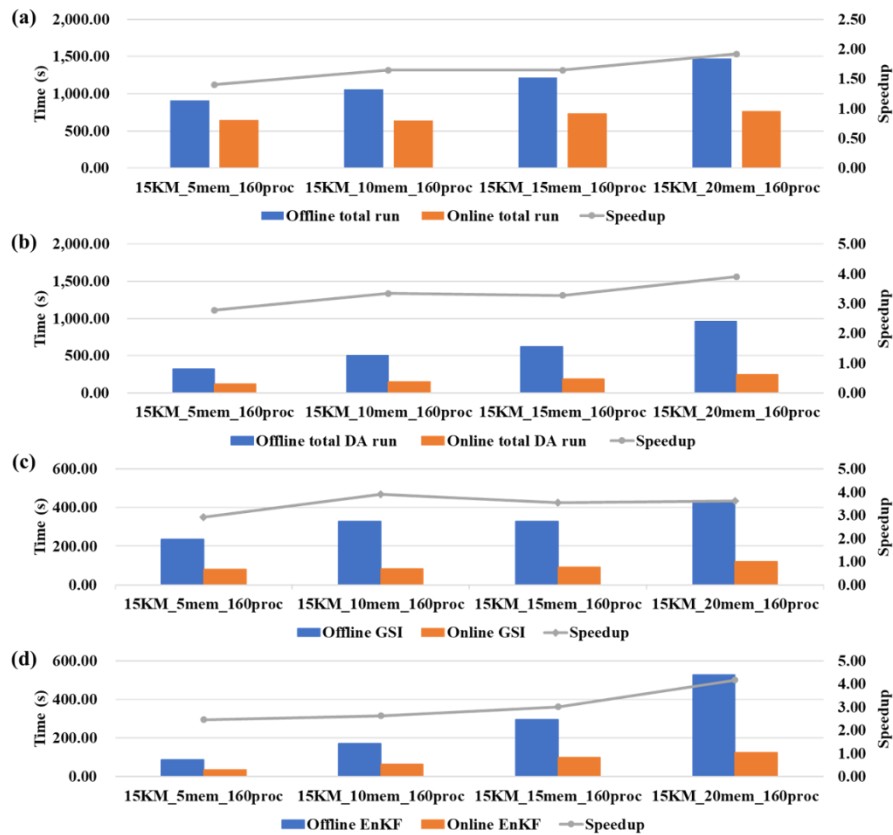

**Figure 11. Execution time (colored bars) corresponding to the online and offline WRF-GSI/EnKF and the corresponding speedup (gray line, ratio of offline execution time to online execution time) from experiment set 1 in Table 2. (a) Total run (including model run and DA algorithms run). (b) DA algorithms (including GSI and EnKF) run. (c) GSI run. (d) EnKF run.**



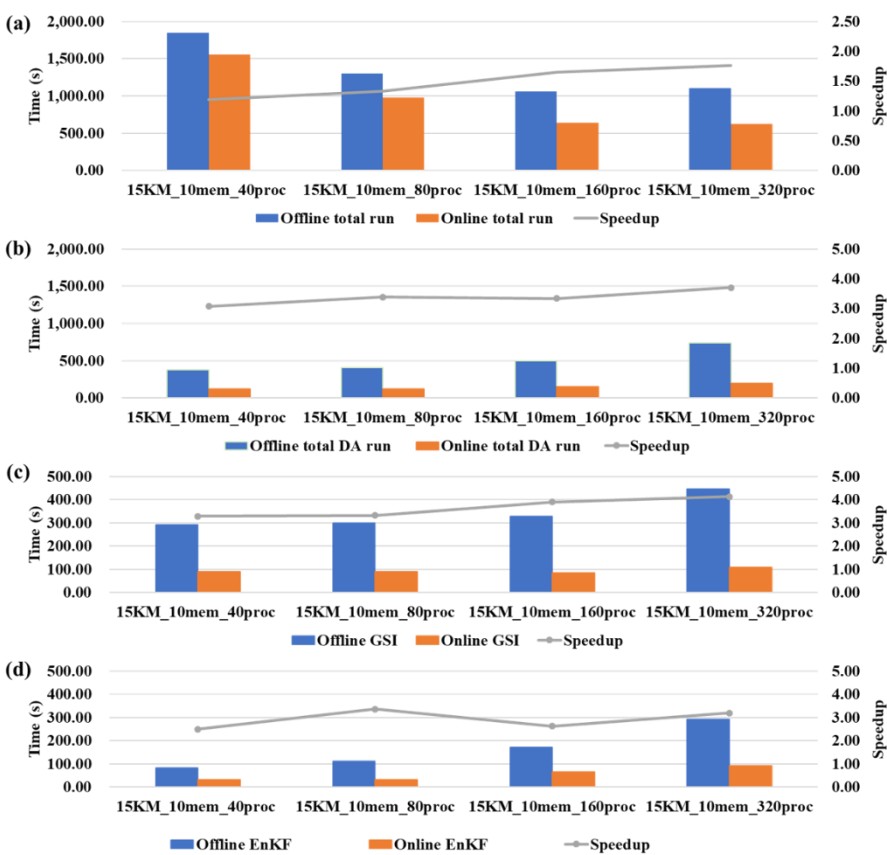

**Figure 12. As in Fig. 11, but from experiment set 2 in Table 2.**





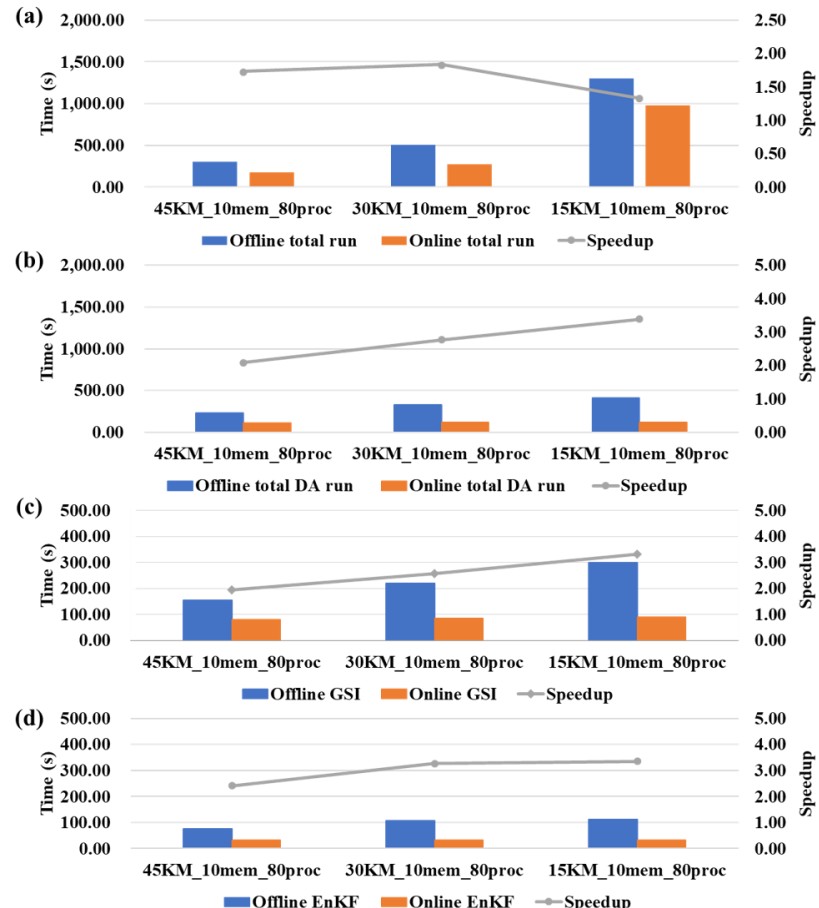

**Figure 13. As in Fig. 11, but from experiment set 3 in Table 2.**

...



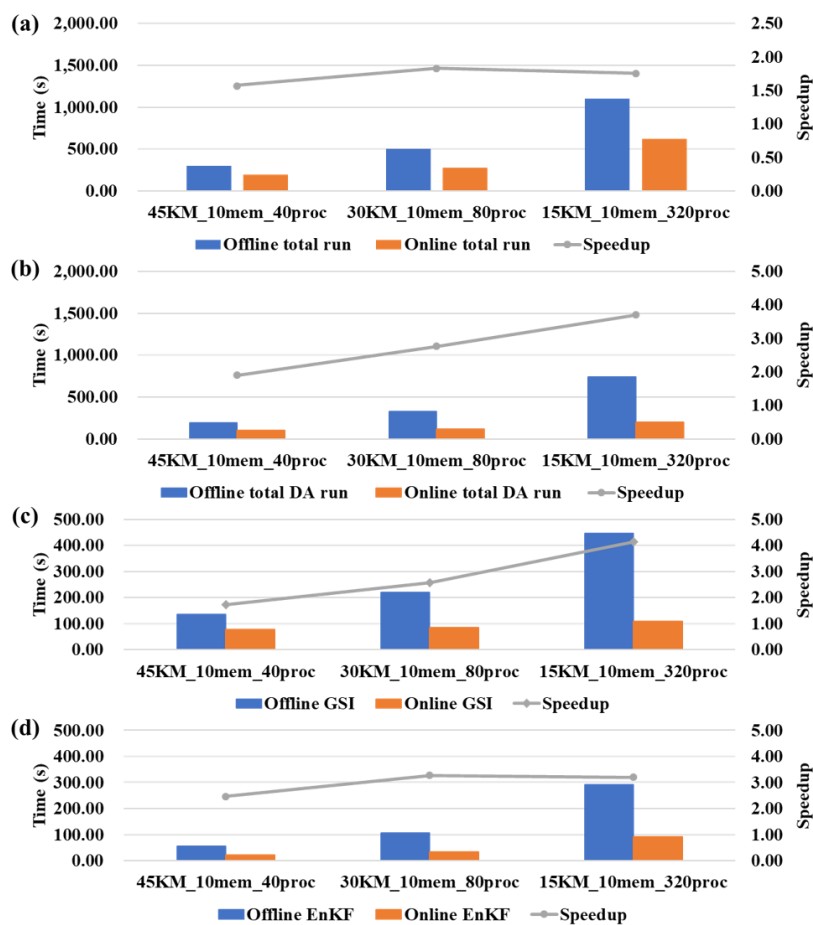


**Figure 14. As in Fig. 11, but from experiment set 4 in Table 2.**

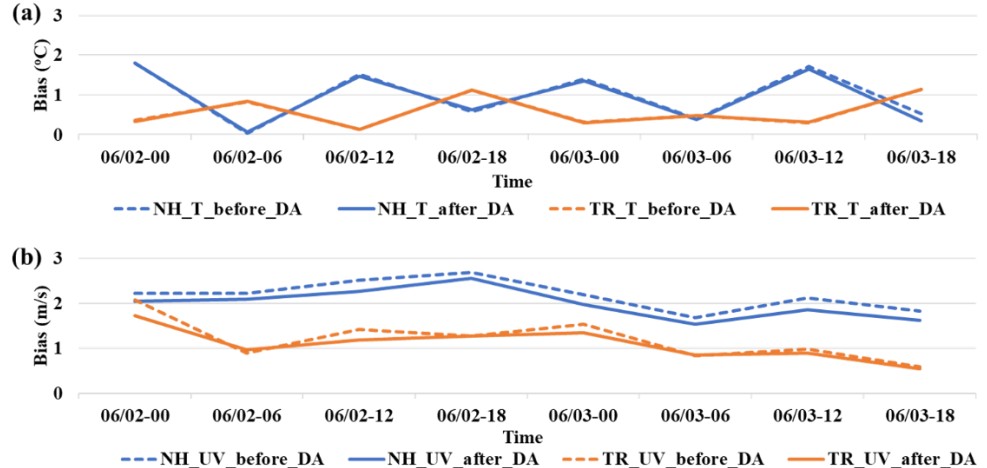

**Figure 15. Total bias of assimilated variables relative to corresponding observations before and after DA for (a) T and (b) UV at each DA time from the EnKF standard output file. The dotted lines indicate the bias of assimilated variables before DA and the solid lines indicate the bias of assimilated variables after DA. Blue lines are the bias in the tropics (25° S–25° N), and orange lines are the bias in the Northern Hemisphere area (25°–90° N).**





**Table 1. Horizontal resolutions and time steps of WRF.**

| Horizontal Resolution | Total Horizontal Grid Points | Time Step |
|---|---|---|
| 45 km | 160×120 | 180 s |
| 30 km | 240×180 | 120 s |
| 15 km | 480×360 | 60 s |

**Table 2. Setup of four experiment sets in terms of horizontal resolution, number of ensemble members and number of processes**

| Experiment set | Horizontal resolution | Number of ensemble members | Processes for each ensemble member | Label marks |
|---|---|---|---|---|
| **Set 1** | 15 km | 5 | 160 | 15KM_5mem_160proc |
| | | 10 | | 15KM_10mem_160proc |
| | | 15 | | 15KM_15mem_160proc |
| | | 20 | | 15KM_20mem_160proc |
| **Set 2** | 15 km | 10 | 40 | 15KM_10mem_40proc |
| | | | 80 | 15KM_10mem_80proc |
| | | | 160 | 15KM_10mem_160proc |
| | | | 320 | 15KM_10mem_320proc |
| **Set 3** | 45 km | 10 | 80 | 45KM_10mem_80proc |
| | 30 km | | | 30KM_10mem_80proc |
| | 15 km | | | 15KM_10mem_80proc |
| **Set 4** | 45 km | 10 | 40 | 45KM_10mem_40proc |
| | 30 km | | 80 | 30KM_10mem_80proc |
| | 15 km | | 320 | 15KM_10mem_320proc |






**Table 3. I/O access statistics corresponding to WRF-GSI/EnKF**

| Horizontal resolution | Number of ensemble members | Number of observation prior files | Total I/O accesses to observation priors | Number of model ensemble background & analysis files | Total I/O accesses to model ensemble background & analysis files |
|---|---|---|---|---|---|
| 15 km | 5 | 12 | 0.11 GB | 324 | 129.13 GB |
| 15 km | 10 | 22 | 0.21 GB | 624 | 251.30 GB |
| 15 km | 15 | 32 | 0.30 GB | 924 | 373.48 GB |
| 15 km | 20 | 42 | 0.39 GB | 1224 | 495.65 GB |
| 30 km | 10 | 22 | 0.18 GB | 624 | 62.86 GB |
| 45 km | 10 | 22 | 0.17 GB | 624 | 27.96 GB |