# Peer review of "Developing a common, flexible and efficient framework for weakly coupled ensemble data assimilation based on C-Coupler2.0"

_Geoscientific Model Development, 2020_

## Short Comment (SC1) · 29 May 2020

Dear authors,

thank you for this interesting article on the assimilation framework based on C-Coupler.

I am Lars Nerger from the Alfred Wegener Institute in Germany and leading the development of PDAF, the Parallel Data Assimilation Framework. I'm happy to see that our work on PDAF motivated your work so much.

Having read the article, I have the impression that there are some misunderstandings about the functionality and requirements of PDAF. Apparently not all aspects have

been sufficiently clarified in the response of my coauthors and me to Dr. Liu's comment on our article "Efficient ensemble data assimilation for coupled models with the Parallel Data Assimilation Framework: Example of AWI-CM" (https://doi.org/10.5194/gmd-2019-167). Thus, I like to clarify these points so that they can be correctly described in your article:

**In lines 84-86 you write** *"PDAF addressed the problem of generating local communicators by imposing a precondition of process layout such that each ensemble member uses the same number of processes with successive IDs in the MPI_COMM_WORLD."*
PDAF does actually not require this precondition of process layout. It is not required that the processes have successive IDs in MPI_COMM_WORLD. With PDAF we provide the routine init_parallel_pdaf to perform the initialization of the MPI communicators. This is a template file, which can be adapted to the users' needs. The default configuration uses successive IDs, because it's simple to implement and consistent with most process layouts used in the models. However, this can be changed by the user, and an arbitrary ordering of the IDs would be possible (for typical cases like ocean or atmospheric models this would hardly be useful, but we cannot exclude that there are models which benefit from such distribution of the processes). It is also not strictly required that each ensemble member uses the same number of processes. This is the default setup, which should be a reasonable choice as running different ensemble members with different numbers of processes likely results in different execution times and hence some processes would need to wait for others when it comes to the analysis step of the ensemble data assimilation.

**In lines 86-88 you write** *"For example, if there are 10 members in an ensemble run and each ensemble member uses 50 processes, there should be 500 processes (#0-#499) in the MPI_COMM_WORLD, and processes #0-#49 will be used for the first ensemble member"*
It follows from the previous clarification that the statement on processes #0-#49 only

holds for the default setup, but not in general. Further, it is not required that there are 500 processes, thus enough processes to integrate all ensemble states in parallel. PDAF provides the flexible parallelization variant, in which, e.g., the 10 members of the example could be run using 250 processes, or even only 50 processes. This parallelization variant is described in the PDAF documentation on the PDAF web site http://pdaf.awi.de and in Nerger et al. (2005).

**In lines 88-90 you write** *"Such a solution can facilitate the corresponding code implementation, but requires the user to know and guarantee the process number precondition when submitting the MPI job of an ensemble run."*
As described before, with PDAF one does not need the full amount of processes. Of course one does need to know the number of processes and needs to ensure this number when submitting an MPI job. This is the case for any MPI execution since one needs to specify the number of processes in the call to mpirun (or mpiexec/aprun/srun) and for supercomputers with batch systems one needs to know this number when specifying the resource requirements for a compute job.

**In lines 98-100 you write** *"... the user is still required to perform many tasks; e.g., developing the call back functions for gathering the same model field from all ensemble members into a vector of the DA method and distributing the data of a field in a vector to all ensemble members"*
It is correct that such operations are required. However, implementing them is a pretty trivial task. Actually, this operation is always model-specific and an analogous operation even happens when using a coupler like C-Coupler. However, in this case it's in the coupler API instead of the assimilation API.

**In lines 101-103 you write** *"The user is recommended to make the DA implementation using the same parallel decomposition (grid domain decomposition for parallelization) as the model, and needs to develop new code when the model and DA implementation use different parallel decompositions."*
In deed we recommend to use the same domain decomposition. To our experience this
makes it easier for the user to implement the call-back routines. Nonetheless, PDAF allows a user to use more processes for the analysis step, but then the user has to ensure that the model fields are properly distributed.

**In lines 103-104 you write** *"... efforts should be made to enable the software compilation system of the model to compile the code of the DA methods."*
This is actually not correct. One usually compiles the PDAF core library, which contains the code of the DA methods, separately and later links this library with the model code. Here, it's not relevant if the PDAF library is compiled as a static or dynamically linked library (This seems to be rather a matter of personal taste, even though for some supercomputers like the Cray XC series the use of dynamically linked libraries is not recommended). For the user-code (the call-back routines), we recommend to compile the code with the model. However, this is not strictly required and only could also compile this code separately as a library. Anyway to our experience it's easier to compile the code with the model, because it helps to ensure a consistent compilation.

**In lines 108-110 you write** *"Although PDAF enables a DA algorithm to run in parallel, it only makes the processor cores of the first ensemble member available to the DA algorithm and forces the processor cores used by other ensemble members to idle when running the DA algorithm."*
As described above in relation to lines 86-88 and lines 101-103, it is not correct that PDAF "only makes the processor cores of the first ensemble member available to the DA algorithm". This is just the default configuration, which can be adapted. However, to our experience the analysis step, i.e. the execution of the DA algorithm, is usually fast compared to the time to compute the ensemble forecast. Thus by letting more processes compute the DA algorithm, the overall speedup will be limited. In contrast, using more processes for the analysis step requires remapping of the domain decompositions. This requires more MPI communication calls, which will take more time than just collecting ensemble information on the first ensemble task. Thus, while one gains speed in the analysis one looses time in the remapping. What is faster will

be case-dependent.

I hope that the explanations above help to correct the statements about PDAF in your manuscript. Overall, I have the impression that one has to be very careful when claiming limitations because the risk to be not fully accurate is high. From reading your article I have the impression that it rather presents a different strategy to couple the DA algorithm with the model, than one that overcomes limitations. PDAF provides a lightweight coupler, particularly aimed at data assimilation, which can be easily connected to a model, even if the model already uses a model-coupler (like OASIS, which is implemented in the AWI-CM model used by our study mentioned above). In the framework described in your article a complex coupler is used, which can, e.g., perform mapping between different domain decompositions. However, using this coupler will e.g. require to implement subroutine calls that describe the model grid so that this information can be handled by the coupler and the by DA algorithm (When I read this correctly in the source code you provided with the article, this is done in module_wrf_DA.F which contains about 56 calls to procedures of the coupler). Thus, it seems to be more work to add the coupler functionality of C-Coupler to a model than just writing model fields into a vector as is done in PDAF.

Overall, as the limitations of PDAF, which are claimed in the article, do actually not exist (all three bullets of the list at the beginning of Section 3 are also fulfilled by PDAF), I would find it only suitable to describe the framework based on C-Coupler as a strategy that is 'different' from that of PDAF, rather than a strategy that solves apparent limitations.

Kind regards,
Lars Nerger

---

## Short Comment (SC2) · 30 May 2020

Dear Dr Lars Nerger,

Thanks a lot for your introductions, comments and suggestions.

We are sorry about the misunderstandings regarding PDAF in the manuscript. We will correct them when revising the manuscript based on more discussions with you.

Here, I'd like to briefly introduce the background about DAFCC and this manuscript. We began to design and develop DAFCC in 2017 according to the requirements from operational model development in China. We noted your PDAF work after someone

introduced it to us in 2018. Then we know that PDAF is a pioneer of our work especially after your latest GMDD manuscript was online. To make DAFCC known by potential users and to share some "new" aspects in developing a DA framework, we submitted this manuscript to GMD for possible publication. We had to state the differences between PDAF and DAFCC because it will be easily asked what are new advancements from the state of the art in the review process.

The statements about PDAF in the manuscript are from our understandings based on the documentations, papers and existing code versions (including the latest public version V1.15.1) of PDAF. It is a wrong statement that PDAF requires each ensemble member use the processes with successive IDs in the MPI_COMM_WORLD, because the communicators COMM_filter, COMM_model and COMM_couple are generated by the corresponding user code. We will correct this statement when revising the manuscript. There would be differences between PDAF and DAFCC regarding the generation of communicators, where PDAF relies on users' efforts while DAFCC automatically generates communicators implicitly. According to our experiences from the cooperation with Chinese model teams, we just feel that, it is not easy to develop the codes for generating the communicator of the ensemble of a component model in the ensemble run of a coupled model, especially for scientists.

We inferred that PDAF requires all model ensemble members use the same number of processes and the same parallel decomposition, and only makes the processor cores of the first ensemble member available to the DA algorithm, based on the PDAF codes (e.g., version V1.15.1). For example, in the "SUBROUTINE PDAF_get_state" in the PDAF code file "PDAF-D_get_state.F90", the root process in COMM_couple corresponding to one ensemble member gets state variables from the remaining processes in COMM_couple corresponding to other ensemble members. Based on the examples available in the code package, we know that the global communicator is generally split into a set of COMM_couple each of which corresponds to the ith process of all ensemble members. We really do not know how to organize COMM_couple and whether

"PDAF_get_state" still works, under the case that model ensemble members use different numbers of processes or different parallel decompositions, or a DA algorithm uses all processes of the ensemble. Could you please show some examples about that case? Thanks.

We are sorry of that "... efforts should be made to enable the software compilation system of the model to compile the code of the DA methods." is incorrect. How about "... efforts should be made to enable the software compilation system of PDAF to compile the code of the DA methods."

Regarding your comments "However, implementing them is a pretty trivial task. Actually, this operation is always model-specific and an analogous operation even happens when using a coupler like C-Coupler. However, in this case it's in the coupler API instead of the assimilation API.". In our opinion, it will be not a pretty trivial task but a heavy task even like developing a new coupler when the DA algorithm uses a different parallel decomposition. C-Coupler has formalized model-specific operations for data transfer among different component models or different parallel decompositions with standard APIs, like other couplers. So DAFCC that is based on C-Coupler2 does not require users to conduct such tasks.

Regarding your comments "Thus by letting more processes compute the DA algorithm, the overall speedup will be limited. In contrast, using more processes for the analysis step requires remapping of the domain decompositions. This requires more MPI communication calls, which will take more time than just collecting ensemble information on the first ensemble task. Thus, while one gains speed in the analysis one looses time in the remapping. What is faster will be case-dependent.". We agree that what is faster will be case-dependent. That's why we try to offer a maximum number of processes to DA algorithms while a DA algorithm can only use a part of processes it can effectively use in real cases. Could you please show us a detail example how PDAF enables a DA algorithm to use all processes in the ensemble of a component model (the DA algorithm will generally use a very different parallel decomposition from the

model ensemble), without modifying the PDAF codes (e.g., version V1.15.1) and with trivial efforts in developing call back functions.

Wish more discussions with you. Then we can list out the new statements regarding PDAF. Many thanks again.

Best regards,

Li Liu

---

## Short Comment (SC3) · 15 Jun 2020

**Dear Dr. Liu,**

Dear Dr Lars Nerger, Thanks a lot for your introductions, comments and suggestions.

We are sorry about the misunderstandings regarding PDAF in the manuscript. We will correct them when revising the manuscript based on more discussions with you.

**My time to perform this discussion is unfortunately very limited. The essential information to correct your statements about PDAF is already in my initial comment. Thus, I will keep this second comment short.**

[Figure]

Here, I'd like to briefly introduce the background about DAFCC and this manuscript. We began to design and develop DAFCC in 2017 according to the requirements from operational model development in China. We noted your PDAF work after someone introduced it to us in 2018. Then we know that PDAF is a pioneer of our work especially after your latest GMDD manuscript was online. To make DAFCC known by potential users and to share some "new" aspects in developing a DA framework, we submitted this manuscript to GMD for possible publication. We had to state the differences between PDAF and DAFCC because it will be easily asked what are new advancements from the state of the art in the review process.

**Well, claiming that the system you describe solves apparent limitations of PDAF, which as I explained is incorrect, is an interesting strategy when you want to point out "new aspects". Obviously, there is also the significant risk to be inaccurate. To my impression there are enough differences in the strategy to use a complex coupler compared to the approach of PDAF to by-pass the coupler. This difference is also in the situation that PDAF is designed to be compatible with any model coupler (even C-Coupler), while one would need to figure out how to use the C-Coupler based system if a coupled model is already implemented with a coupler like OASIS-MCT. The obvious advancement I see is that when one has a coupled model that already uses C-Coupler, it might be easier to add the data assimilation functionality - at least if there is a data assimilation algorithm already coded for the model as in the example in your article.**

There would be differences between PDAF and DAFCC regarding the generation of communicators, where PDAF relies on users' efforts while DAFCC automatically generates communicators implicitly. According to our experiences from the cooperation with Chinese model teams, we just feel that, it is not easy to develop the codes for generating the communicator of the ensemble of a component model in the ensemble run of a coupled model, especially for scientists.

**This experience doesn't coincide with ours. The templates for the communicator**

**setup that we provide with PDAF are readily usable. Thus, there is no particular "users' effort".**

We inferred that PDAF requires all model ensemble members use the same number of processes and the same parallel decomposition, and only makes the processor cores of the first ensemble member available to the DA algorithm, based on the PDAF codes (e.g., version V1.15.1). For example, in the "SUBROUTINE PDAF_get_state" in the PDAF code file "PDAF-D_get_state.F90", the root process in COMM_couple corresponding to one ensemble member gets state variables from the remaining processes in COMM_couple corresponding to other ensemble members. Based on the examples available in the code package, we know that the global communicator is generally split into a set of COMM_couple each of which corresponds to the ith process of all ensemble members. We really do not know how to organize COMM_couple and whether "PDAF_get_state" still works, under the case that model ensemble members use different numbers of processes or different parallel decompositions, or a DA algorithm uses all processes of the ensemble. Could you please show some examples about that case? Thanks.

**The key is in deed in the setup of COMM_couple. I think this case is analogous to that of letting all processes perform the analysis step, which is discussed further below. (Nonetheless, I'm not aware of any situation where a different decomposition would be useful for an ensemble of equivalent model states where one does not know before the actual computation that any of them would be particularly faster or slower than the others)**

We are sorry of that "... efforts should be made to enable the software compilation system of the model to compile the code of the DA methods." is incorrect. How about "... efforts should be made to enable the software compilation system of PDAF to compile the code of the DA methods."

**If you also intend to write " ... efforts should be made to enable the software**

**compilation system of C-Coupler to compile" your alternative sentence would make sense. Obviously, also the code of C-Coupler and the assimilation code you use need to be compiled. Thus, it might be even more effort in your case since C-Coupler and the assimilation codes are really separate.**

Regarding your comments "However, implementing them is a pretty trivial task. Actually, this operation is always model-specific and an analogous operation even happens when using a coupler like C-Coupler. However, in this case it's in the coupler API instead of the assimilation API.". In our opinion, it will be not a pretty trivial task but a heavy task even like developing a new coupler when the DA algorithm uses a different parallel decomposition. C-Coupler has formalized model-specific operations for data transfer among different component models or different parallel decompositions with standard APIs, like other couplers. So DAFCC that is based on C-Coupler2 does not require users to conduct such tasks.

**With PDAF this is not at all a 'heavy task'. Let me show this on two examples which are in the model binding codes provided by the PDAF package:**

**For the ocean model FESOM in AWI-CM, we use lines like**

```
DO i = 1, myDim_nod3D
    state_p(i + offset(2)) = uf(i)
END DO
```

**which writes the meridional velocity into the state vector. Analogously, for the ocean model MITgcm we use**

```
    DO bj=myByLo(myThid),myByHi(myThid)
       DO bi=myBxLo(myThid),myBxHi(myThid)
          DO k=1,Nr
             DO j=1,sNy
```

```
                    DO i=1,sNx
                        koffset = koffset + 1
                        state_p(koffset) = uVel(i,j,k,bi,bj)
                    ENDDO
                ENDDO
            ENDDO
        ENDDO
    ENDDO
```

**where the two outermost loops provide support for the threading in MITgcm and the other loops run simply over the three indices of this 3D field. Actually, we have done such implementations for various models and the scheme is always the same. For me, that's 'pretty trivial'.**

Regarding your comments "Thus by letting more processes compute the DA algorithm, the overall speedup will be limited. In contrast, using more processes for the analysis step requires remapping of the domain decompositions. This requires more MPI communication calls, which will take more time than just collecting ensemble information on the first ensemble task. Thus, while one gains speed in the analysis one looses time in the remapping. What is faster will be case-dependent.". We agree that what is faster will be case-dependent. That's why we try to offer a maximum number of processes to DA algorithms while a DA algorithm can only use a part of processes it can effectively use in real cases. Could you please show us a detail example how PDAF enables a DA algorithm to use all processes in the ensemble of a component model(the DA algorithm will generally use a very different parallel decomposition from the model ensemble), without modifying the PDAF codes (e.g., version V1.15.1) and with trivial efforts in developing call back functions.

**I don't have a detailed example at hand, and don't have time to prepare one for this discussion about correcting statements about PDAF in your article. Anyway, I can shortly describe what's required.**

Since PDAF itself has no remapping functionality small changes to the user code are required. As global filters are not relevant for high-dimensional models, I'm considering here the localized filters like LETKF and LESTKF. In these filters, PDAF has a loop running over local analysis domains, and a finer domain decomposition will shorten these loops. Further, I'm considering here the case that each model subdomain used in the forecast is further decomposed, which should be sufficiently flexible.

In short the modification would be as follows: One modifies the parallel setup so that COMM_couple consists of a single process each, which switches off communication in COMM_couple. Then, in the call-back routines (namely in pre-poststep_pdaf.F90) one would add a subroutine to collect the ensemble on the sub-subdomains of the model (thus, on chunks of the state vector) and to distribute the ensemble states again after the analysis update. Now, one adapts the number of local analysis domains for the local analysis loop and adds an offset for the loop counter when accessing the state vector elements according to the finer domain decomposition. Then, the filter runs over a smaller number of local analysis domains for each process, without further modification of the code. This code version would also be compatible wit the case that the filter is executed on a lower number of precesses or only on the processes of the first model task.

I hope that these additional explanations made my initial comments about the PDAF features even clearer.

Kind regards,
Lars Nerger

---

## Short Comment (SC4) · 18 Jun 2020

Dear Dr. Nerger,

Thanks a lot for taking time to give further clarifications.

We will try to reduce statements or discussions about PDAF and discuss more about the "new aspects" when revising the manuscript, if reviewers agree.

Here we'd like to reply some of your points.

1. To my impression there are enough differences in the strategy to use a complex coupler compared to the approach of PDAF to by-pass the coupler. This difference is

also in the situation that PDAF is designed to be compatible with any model coupler (even C-Coupler), while one would need to iňĄgure out how to use the C-Coupler based system if a coupled model is already implemented with a coupler like OASIS-MCT.

Response: C-Coupler2 has a functionality namely incremental coupling which enables C-Coupler2 to work cooperatively with other couplers. So, an existing coupled model that uses any other coupler can also use DAFCC enclosed in C-Coupler for data assimilation, without modifying the codes regarding model coupling. Thanks a lot for your reminder. We will introduce this point when revising the manuscript.

2. This experience doesn't coincide with ours. The templates for the communicator setup that we provide with PDAF are readily usable. Thus, there is no particular "users' effort".

Response: We do believe that these templates can work in most cases, while we note that they should be further adapted for special cases (you may have stated this point in your first round of comment), e.g., the distribution of process IDs among ensemble members is irregular.

3. Nonetheless, I'm not aware of any situation where a different decomposition would be useful for an ensemble of equivalent model states where one does not know before the actual computation that any of them would be particularly faster or slower than the others.

Response: We agree that it would be strange if different ensemble members use different parallel decomposition. For example, would it be a real case that DA is performed for an ensemble of WRF on the same big domain, while only one ensemble member has nested domains? Under such a case, the unique ensemble member with nested domains should use more processor cores so as a different parallel decomposition on the big domain for accelerating integration. As we cannot accurately assert the requirements arising from future model developments, we have to make DAFCC as common

Interactive
comment

as possible.

4. If you also intend to write " ... efforts should be made to enable the software compilation system of C-Coupler to compile" your alternative sentence would make sense. Obviously, also the code of C-Coupler and the assimilation code you use need to be compiled. Thus, it might be even more effort in your case since C-Coupler and the assimilation codes are really separate.

Response: We do not require the software compilation system of C-Coupler to compile a DA algorithm because dynamic linking is used in DAFCC. When there are a lot of DA algorithms but only using one, only one DA algorithm can be compiled. When integrating a new DA algorithm, the C-Coupler code and mode code can be unchanged. Compilation is one of our consideration while it may not be a critical point when using a framework. We will modify the manuscript accordingly.

5. In short the modiïfication would be as follows: One modiïfies the parallel setup so that COMM_couple consists of a single process each, which switches off communication in COMM_couple. Then, in the call-back routines (namely in pre-poststep_pdaf.F90) one would add a subroutine to collect the ensemble on the sub-subdomains of the model (thus, on chunks of the state vector) and to distribute the ensemble states again after the analysis update.

Response: Could you please show us the benefit of using PDAF under such a case? Thanks. As the required call-back routines function similarly to the data exchange functionality in DAFCC that is based on C-Coupler, it would be a very challenging work for users to develop such call-back routines. Moreover, further efforts would be required to generate COMM_filter that combines all processes of the whole ensemble of a component model.

Wish more discussions with you again.

Many thanks again.

[Figure]

Best regards,

Li

---

## Referee Comment (RC1) · Anonymous Referee #1 · 29 Jun 2020

The present paper reports the development of a framework for a coupled data assimilation (CDA). Works related to CDA require important technical preparation where many obstacles may occur. This might be related to numerous problems ranging from the compilation of a coupled model code along with a coupler code, communications between different models to purely IT-related problems like MPI libraries and so on. All these problems have to be addressed prior to scientific problems of choosing an optimal coupling scheme between models, and finally the CDA problems like accurate parametrization of error covariance matrices in a coupled framework etc. That is why the present work is a necessary first step towards a CDA system. And only technical aspects of the development are discussed here. However, I suggest that the authors

work more on the paper text. The current text is difficult to understand. The authors present their system as a framework towards future CDA systems describing the system using words like a DA system, a numerical model etc. Conversely, the PDAF system is mentioned, which is somehow used along with the existing DA system that authors call GSI/EnKF. But the use of PDAF may be limited to models that can be incorporated inside the PDAF Fortran code and called as subroutines which is not the case for every model. That is why the authors might clarify early in the text what is the configuration of their future CDA system and what are the purposes of the system.

These are my specific comments.

First, please increase the dpi for the figures, it's hard to read them.

Line 21: please properly introduce the acronyms: "…system GSI/EnKF and the coupled model FIO-AOW".

Lines 26-31: Here you are talking about two widely used DA algorithms: ensemble techniques derived from the Kalman filter and variational data assimilation. But there are methods that you don't mention. Please reformulate the paragraph accordingly. Besides, I would suggest to remove the sentence ", can be viewed as a special case of ensemble-based methods with only one member in the ensemble when we attempt to design and develop a software framework for data assimilation." (lines 30-32).

Line 34: Please add the Environment and Climate Change Canada's (ECCC) hybrid 4D-EnVar DA used operationally in the weather prediction system: Buehner, M., Mc-Taggart‐Cowan, R., Beaulne, A., Charette, C., Garand, L., Heilliette, S., Lapalme, E., Laroche, S., Macpherson, S.R., Morneau, J. and Zadra, A. (2015) Implementation of deterministic weather forecasting systems based on ensemble‐variational data assimilation at Environment Canada. Part I: the global system. Monthly Weather Review, 143, 2532–2559

Lines 36-37: With the rapid development of science and technology, numerical forecasting systems with only an individual component model (such as an atmospheric model) have reached a predictability limit. Please reformulate it or add appropriate references.

Lines 37-38: Coupled models have been widely used in numerical forecasting to break the bottleneck of the limited predictability... I would also reformulate it in a way that it was established that coupled models may provide better results with respect to uncoupled models. Besides, you may wish to add references for coupled models that used operationally in several operational centres (Met Office, ECMWF, ECCC, etc.)

Lines 42-43: add references to coupled data assimilation (CDA) systems developed in JAMSTEC, NOAA, Met Office, ECMWF, ECCC: Sugiura et al., 2008, Mochizuki et al., 2016, Yang et al., 2013, Zhang et al., 2014, Lea et al., 2015, Laloyaux et al. 2016 and 2018, Browne et al., 2019, Skachko et al. 2019

Lines 45-49: It is not clear what do you mean. Reformulate or simply remove this sentence.

Line 49: You don't mention the NASA GMAO system, for example. I would suggest changing "most" by "several".

Lines 49-60: The ensemble size of current global operational systems used in NWP generally exceeds 256 members. The global models used in the ensemble are run on grids with horizontal sampling of several tens of kilometers. So, the I/O operations are currently necessary. If the aim of the paper is not to discuss real systems, please state it more clearly in the abstract and the beginning of introduction. It will help to understand what kind of CDA system you are developing and its future use.

Line 62-63: reformulate the meaning of WCDA. Not the models are assimilated independently, but data from model components are assimilated independently by two separate DA systems.

Line 76-78: It's obviously more efficient. However, you don't mention the feasibility of

this approach when a model (or several models and couplers) is called in the PDAF Fortran code is a subroutine.

Line 332: again, GSI/EnKF has not been defined yet. What is GSI?

Lines 330-343: It's difficult to understand this section. First, you have an atmosphere-ocean-wave coupled model. Second, you have a DA system that you call GSI/EnKF. Does this system compute atmospheric analyses? How ocean analyses are computed? Do you compute also analysis for the waves?

Line 345: What do you mean by "combines"? Is it a hybrid system? Or you may choose between two options: a variational (which one?) and ensemble technique? You may also state explicitly that both options share parts of codes, for example, observation operators.

Lines 345-350: What DA method is used in your experiments: EnKF, EnSRF or ETKF? Please clarify.

Instead of using "pure ensemble DA", you may simply say that the EnKF (or ETKF or whatever) mode was chosen.

Line 354-371: atmospheric DA? So, do you use EnKF to perform atmospheric analyses?

L373: what step you are talking about? An atmospheric model time step?

l.374-395: As shown. . . It's not clear.

L397: where it was used? And who has upgraded the coupled model to the c-coupler 2.0? After I read the section 5, I realized that there is no ocean data assimilation in your system. So I wonder why you call your prototype WCDA? Your system is an atmospheric DA using a coupled model to compute background states.

---

## Short Comment (SC5) · 29 Jun 2020

**Dear Dr. Liu,**

Dear Dr. Nerger,

Thanks a lot for taking time to give further clarifications.

We will try to reduce statements or discussions about PDAF and discuss more about the "new aspects" when revising the manuscript, if reviewers agree.

**In deed, I hope that my comments will help you, the reviewers and the editor to ensure that no incorrect claims about PDAF will be included in the final**

**manuscript. They hopefully also help to avoid exaggerations regarding the complexity of tasks like the process setup, the compilation aspect, or adaptions for using all processes in the analysis step.**

Here we'd like to reply some of your points.

2: We do believe that these templates can work in most cases, while we note that they should be further adapted for special cases (you may have stated this point in your first round of comment), e.g., the distribution of process IDs among ensemble members is irregular.

**It's a design philosophy of PDAF to provide templates that are readily usable for the typical cases, and flexibility to adapt for special cases. This allows us to keep the implementation aspects particularly easy for the typical cases - which are the vast majority by definition of 'typical'.**

3: We agree that it would be strange if different ensemble members use different parallel decomposition. For example, would it be a real case that DA is performed for an ensemble of WRF on the same big domain, while only one ensemble member has nested domains? Under such a case, the unique ensemble member with nested domains should use more processor cores so as a different parallel decomposition on the big domain for accelerating integration. As we cannot accurately assert the requirements arising from future model developments, we have to make DAFCC as common as possible.

**For ensemble filters one always needs a set of equivalent ensemble members to estimate the ensemble spread or covariance matrix. Thus, the case with only a single ensemble member with nesting is not usable.**

4: We do not require the software compilation system of C-Coupler to compile a DA algorithm because dynamic linking is used in DAFCC. When there are a lot of DA algorithms but only using one, only one DA algorithm can be compiled. When integrating

a new DA algorithm, the C-Coupler code and mode code can be unchanged. Compilation is one of our consideration while it may not be a critical point when using a framework. We will modify the manuscript accordingly.

**Well, your previously suggested statement "... efforts should be made to enable the software compilation system of PDAF to compile the code of the DA methods." is incorrect, because the DA methods are part of PDAF and the claimed effort does simply not exist. When you check the documentation of PDAF, you will see that the compilation is one aspect where we provide recommendations that make this task particularly simple.**

5. In short the modification would be as follows: One modifies the parallel setup so that COMM_couple consists of a single process each, which switches off communication in COMM_couple. Then, in the call-back routines (namely in pre-poststep_pdaf.F90) one would add a subroutine to collect the ensemble on the sub- subdomains of the model (thus, on chunks of the state vector) and to distribute the ensemble states again after the analysis update.
Response: Could you please show us the benefit of using PDAF under such a case? Thanks. As the required call-back routines function similarly to the data exchange functionality in DAFCC that is based on C-Coupler, it would be a very challenging work for users to develop such call-back routines. Moreover, further efforts would be required to generate COMM_filter that combines all processes of the whole ensemble of a component model.

**The obvious benefit is that PDAF provides the parallelized DA methods. Further, PDAF takes care of the ensemble and the state vectors. Moreover one still has the flexibility of using a different number of processes for the analysis step - the adapted code that I described in my previous comment can also be used for an analysis step using e.g. only the processes of a single model task.**
**The required call-back routine does actually not need a functionality 'similarly to the data exchange functionality in DAFCC'. For the DA one does not need to**
**develop an actual domain decomposition, but one distributes the state vector that corresponds to a process domain. This approach is much easier than the general domain remapping that one needs in a coupler.**
**COMM_filter is just the full set of processes. There are no 'further efforts' to generate this (in init_parallel_pdaf.F90 one only needs to change two lines of code and remove one if-clause to obtain the changed COMM_coupled and COMM_filter that I described in my previous comment).**

**Kind regards,**
**Lars Nerger**

---

## Short Comment (SC6) · 30 Jun 2020

Dear Dr. Nerger,

Thanks a lot for further discussions.

We still have some questions as follows.

1. About the conception and target regarding the term of "framework"

Response: after reading your statements, we feel that we have a different conception and goal regarding the term of "framework" from you. In our mind, a framework is more about software infrastructure rather than science. It is generally a common middleware

that effectively serves the combination between different scientific modules. Although a user can benefit from the existing DA algorithms that have been adapted to DAFCC by other users (if they'd like to share their contributions), we view it as a common benefit of using a framework but not a specific benefit of DAFCC, because a framework that generally formats a set of interfaces will improve shareability. For specific benefits, we try to make DAFCC as common, efficient and convenient as possible. Although DAFCC does not include and is not bound to DA algorithms that are external procedures in dynamic-linking libraries, we believe that various DA algorithms can be easily adapted to DAFCC, including existing DA methods in PDAF. Moreover, we view a combination DAFCC with specific models and specific DA algorithms as a platform. We sincerely welcome any user to use DAFCC to develop a such platform. According to your statements, PDAF seems more about a platform in our mind, as it includes specific DA methods based on your direct contributions to both science and software engineering. Under such a case, commonality and convenience for integrating a DA method into PDAF are not critical to users of PDAF. Regarding the parallelism of DA algorithms, we still believe that higher parallelism generally means the opportunity to achieve better acceleration. Given that the whole ensemble of 20 members use 10,000 cores, DAFCC conveniently enables a DA method to use a wide range of 1~10,000 cores but not the 500 cores corresponding to a member. This can enable users to find and then use an optimal core number for best acceleration of a DA algorithm.

Now, we tend to think that, we and you have different goals in developing a DA framework. We want to provide freedom and convenience in DAFCC for users to develop a DA system based on their available models and available DA algorithms, while is it your goal to provide the DA methods in PDAF to users?

2. About "For the DA one does not need to develop an actual domain decomposition, but one distributes the state vector that corresponds to a process domain."

Response: Do you mean an implementation based on gather/scatter that is highly inefficient? If not, what the domain decomposition is, and could you please give an

example?

3. About "COMM_filter is just the full set of processes."

Response: Do you mean that COMM_filter is MPI_COMM_WORLD? For example, in an ensemble of an air-sea coupled model, given that only the atmosphere model is assimilated, COMM_filter should only cover the processes of the ensemble of the atmosphere component and thus it should not be MPI_COMM_WORLD. Here, we'd like to ask, how to obtain COMM_filter without 'further efforts' under such a case?

Wish more discussions with you again.

Many thanks again.

Best regards,

Li
* * *

---

## Referee Comment (RC2) · Anonymous Referee #2 · 1 Jul 2020

This article describes the approach of extending the C-Coupler2.0 to provide data assimilation capabilities that can work with coupled models. It is motivated by attempts to use PDAF, however I am unsure that the criticisms of PDAF are sufficient motivation for developing the new system. The design of the new system is described, along with detailed discussion of how different components are implemented. An example of the working system is given where the data assimilation system from GSI/EnKF is ported to DAFCC1 and applied to the different components of the coupled model FIO-AOW in a weakly coupled DA context. Results are shown to indicate that the I/O bottlenecks associated with offline implementation are avoided by using MPI instead of file access.

General comments:

The motivation for developing your own system rather than using PDAF I found lacking. There were 2 main points I could find in section 2, namely "PDAF [... imposes] a precondition of process layout such that each ensemble member uses the same number of processes with successive IDs in the MPI_COMM_WORLD" and "[PDAF] only makes the processor cores of the first ensemble member available to the DA algorithm and forces the processor cores used by other ensemble members to idle when running the DA algorithm". The second statement I think is untrue, but Dr Lars Nerger has posted a short comment on PDAF so I trust he will ensure the correctness there. The first point I find to be obscure as I cannot think of a situation where you would not have ensemble members using sequential MPI process IDs. If you have a specific situation where this is the case, please elaborate on it so the reader can understand why this is important.

One overarching question which is not addressed is why would you design from the outset a "weakly coupled" data assimilation system? Why not design a strongly coupled system and then simplify it? I suppose the answer here is to still be able to piggyback on existing observation processing systems and to allow for different observation frequencies, but this should be clearly set out in the article.

I would like to see more clarity in relation to the comparisons that you make. There are a number of places where the comparison is against an system that uses I/O and reading/writing files from/to disk rather than MPI communications. In such a case phrases like "accelerating the DA system" should be qualified. There are other relations made where it is unclear what the comparison is with. For example in the abstract you state that the new methodology "enables the DA method to utilize more processor cores in parallel execution" but I cannot see the baseline for such a statement. Moreover would such a statement hold with a different baseline?

The article gives a reasonable overview and references for general data assimilation

concepts. However the article should point the reader to some of the latest examples of operational weakly coupled data assimilation. Good references for this include, with the first using PDAF:

Goodliff, M., Bruening, T., Schwichtenberg, F., Li, X., Lindenthal, A., Lorkowski, I., & Nerger, L. (2019). Temperature assimilation into a coastal ocean-biogeochemical model: assessment of weakly and strongly coupled data assimilation. Ocean Dynamics, 69(10), 1217–1237. https://doi.org/10.1007/s10236-019-01299-7

Skachko, S., Buehner, M., Laroche, S., Lapalme, E., Smith, G., Roy, F., . . . Garand, L. (2019). Weakly coupled atmosphere-ocean data assimilation in the Canadian global prediction system (v1). Geoscientific Model Development, 12(12), 5097–5112. https://doi.org/10.5194/gmd-12-5097-2019

Browne, P. A., de Rosnay, P., Zuo, H., Bennett, A., & Dawson, A. (2019). Weakly Coupled Ocean-Atmosphere Data Assimilation in the ECMWF NWP System. Remote Sensing, 11(234), 1–24. https://doi.org/10.3390/rs11030234

Specific comments:

Lines 12,25. "better" than what?

Line 47: "how to conveniently (1) achieve an ensemble run of a coupled model" What is your measure of convenience here? This is a task which is regularly done at many centres around the world, do they all have inconvenient methods for running ensembles of coupled models?

Line 55: On the use of disk files, this is also a robust strategy when it comes to massively parallel computing, as this risk of random task failures increases with the size of the coupled models and the number of ensemble members. This should be noted as positive reason for using disk files, as well as the potential to use a larger ensemble than can be run at a single time on an HPC machine.

Line 57. PDAF is indeed *the* standard for ensemble based DA frameworks. Others

also exist. For example EMPIRE (https://pbrowne.bitbucket.io/empire) Browne, P. A., & Wilson, S. (2015). A simple method for integrating a complex model into an ensemble data assimilation system using MPI. Environmental Modelling \& Software, 68, 122–128. https://doi.org/10.1016/j.envsoft.2015.02.003

Line 93: "How to compile the code of DA methods with the model". This is not necessary. In particular if you run (using MPI) in MPMD mode then the model and the DA could be compiled independently.

Line 108: "Although PDAF enables a DA algorithm to run in parallel, it only makes the processor cores of the first ensemble member available to the DA algorithm and forces the processor cores used by other ensemble members to idle when running the DA algorithm." This is not my understanding of PDAF. I see that Dr Lars Nerger has already submitted comments in relation to PDAF, so I am assured that he will have given you the latest and correct information in relation to this.

You need to discuss other parallel strategies such as that used by P.A. Browne, S. Wilson, 2015.

Line 115: what are such preconditions? Can you give examples where these exist, and if they do, why they are a problem?

Line 122: You should make clear this is because the MPI processes from all ensemble members are available. Or are there even more available?

Line 131: Are you suggesting a coupled model which uses a different coupler, such as OASIS, would then be put into C-Coupler2.0 for the DA component?

Figure 1: This has no explanation. I fail to see the usefulness of this figure.

Line 143: What is the alternative to DLL?

Line 146: "The ensemble component manager is responsible for generating and managing the communicator of ensemble members of a component model." Does this

mean you have a separate ensemble component manager for every component of your coupled model, such as atmosphere, ice, land, composition, etc? If so please state this to help the reader.

Line 155/Figure 3: Is a restriction that the components in each ensemble members run on the same number of MPI processes? Surely there is a restriction enforced by the DA algorithms that the component model is on the same grid for every ensemble member, or has some very exotic DA methodology been implemented? In the case they have, how do you then establish which DA algorithms are applicable given the difference in the ensemble members?

Line 159/160: "execution of a DA algorithm in a component model does not force the processes of other component models to be idled". This must relate to the timestepping procedure of the coupled model. In fact here are you for the first time enforcing that all components of the model must have separate MPI processes? This is not the case in, for example, the ECMWF earth system model (Mogensen, K., Keeley, S. and Towers, P., 2012. Coupling of the NEMO and IFS models in a single executable. Reading, United Kingdom: ECMWF.)

Line 165). Point 4. Please give an examples of such a DA algorithm procedures. Do you mean, for example, that DA algorithm 1 procedure 1 would be calculation of model equivalents ($H(x)$) and DA algorithm 1 procedure 2 would be something like an SVD of the ensemble perturbation matrix?

Line 169: "Scripts are allowed to conduct necessary process control". This comes out of the blue, and it is not clear how this fits within the methodology of having everything using MPI communication. Can you give examples in section 4.4.

Line 188: The "weakly coupled" component of your methodology then relies on using the C-Coupler2.0 to control the coupling of the model then?

Figure 5: Why is there no red within the DA_CCPL_RUN subroutine to indicate data

exchange between the model and the DA?

Section 4.4, 2). Terms such as periodic timer, period_unit, period_count and lag are introduced with no context. These should be defined as well as an explanation of why they are needed for data assimilation.

Section 4.4) I fail to see why any of this is relevant to data assimilation. What is an example of statistical processing in a DA context?

Line 330) "A sample weakly" - do you mean "An example weakly coupled ensemble DA system"? I don't know what "sample" refers to here. This is used many times throughout the manuscript - please clarify.

Section 6.1) The details of the EnSRF (i.e. localization radius and inflation factors) are not useful without a description of the model

Line 410) "We evaluate the effectiveness of DAFCC1 in developing a weakly coupled ensemble DA system". I don't see the justification for this statement. I can see you have implemented the system and shown how it performs computationally with various parameters, as well as a very simplistic verification that the data assimilation is implemented correctly. You should state a measure for effectiveness - was it simply to have a functioning system? Compare this with Browne and Wilson, 2015, where they "propose a simple implementation strategy which does not focus on maximum efficiency of the code. Instead the focus is on the speed of implementation."

Line 424) Why were 3200 cores used when each node has 24 processors? mod(3200,24) != 0.

Line 436) "variables used for DA" -> "are the prognostic, or analysed, variables in the data assimilation".

Section 6.2 could be a simple statement saying that WRF-GSI/EnKF with DAFCC1 is bit-identical to the original offline WRF-GSI/EnKF.

Line 460/Figure 11c) Why does the offline timing of GSI vary with different numbers of ensemble members? On line 458 you state that you run all ensemble members of the offline system concurrently, so I would expect a constant value of time for the model run as you change the number of ensemble members. This clarification will be essential in understanding the rest of the figures here, as otherwise it seems like the comparison may be unfair. Could it be i/o related? With every member trying to write output files at the same time your system slows? If this is the case it should be explicitly accounted for in the final paragraph of this section. Furthermore, you should detail what file system architecture is used at BSCC in section 6.1. Is it something like lustre?

Section 6.4) This is not a measure of the effectiveness in developing a weakly coupled ensemble DA system. Figure 15 shows results for northern hemisphere (25N-90N) and tropics (25S-25N) but I understood this was a limited area system, running from 0N-50N. You should update the figure to reflect this. All this figure appears to show is that the DA system is producing increments which have the correct sign.

---

## Short Comment (SC7) · 2 Jul 2020

Dear Reviewer,

Thanks a lot for reviewing our manuscript and for the comments and suggestions.

We would like to reply some comments here, and will carefully follow all your comments and suggestions when revising the manuscript.

1. About "Conversely, the PDAF system is mentioned, which is somehow used along with the existing DA system that authors call GSI/EnKF. But the use of PDAF may be limited to models that can be incorporated inside the PDAF Fortran code and called

as subroutines which is not the case for every model. That is why the authors might clarify early in the text what is the configuration of their future CDA system and what are the purposes of the system."

Response: we will significantly rewrite about our motivation of DAFCC for future CDA system. Ensemble DA is also developed and used in China. To help the development of ensemble DA in China, especially when model resolution gets finer and DA frequencies get higher, we aim to develop a common ensemble DA framework that can enable users to make DA systems as efficient as possible. As most developers for models and DA systems in China are origin from science and do not have strong experiences in software engineering and parallel programming (many model teams even do not have any full-time software engineer), we have to make DAFCC as convenient as possible, especially for the model developers who are not proficient in parallel programming and parallel debugging with MPI. So, we try to make DAFCC handle as much work as possible. Now, the MPI communicator of whole ensemble of a component model for running an ensemble DA algorithm is generated automatically and then used intra DAFCC and the data exchanges among members, ensemble and DA algorithm are also automatically handled by DAFCC, no matter the differences regarding parallel decompositions. Moreover, we enable a DA algorithm to be enclosed in dynamic-linking library, in order to make the model code and the DA code as independent as possible. GSI/EnKF does not use PDAF currently. Some content in the manuscript may introduce misunderstandings. We will try to correct.

2. About "The ensemble size of current global operational systems used in NWP generally exceeds 256 members. The global models used in the ensemble are run on grids with horizontal sampling of several tens of kilometers. So, the I/O operations are currently necessary. If the aim of the paper is not to discuss real systems, please state it more clearly in the abstract and the beginning of introduction. It will help to understand what kind of CDA system you are developing and its future use."

Response: DAFCC can handle the original I/O operations for data exchange between model ensemble and DA algorithms via MPI, while the I/O operations of operational systems for outputting results can be still kept. Regarding the evaluation in this manuscript, the corresponding I/O operations of WRF as well as other component models are still kept.

Best regards,

Li Liu

---

## Short Comment (SC8) · 4 Jul 2020

Dear Reviewer,

Thanks a lot for reviewing our manuscript and for the comments and suggestions.

We would like to reply some comments here, and will carefully follow all your comments and suggestions when revising the manuscript.

1. The motivation for developing your own system rather than using PDAF I found lacking.

Response: As discussed with Dr Lars Nerger, we have our motivation for developing

DAFCC. We will rewrite the motivation part. Our motivation is as follows (also has been introduced in the reply to the first reviewer). Ensemble DA is also developed and used in China. To help the development of ensemble DA in China, especially when model resolution gets finer and DA frequencies get higher, we aim to develop a common ensemble DA framework that can enable users to make DA systems as efficient as possible. As most developers for models and DA systems in China are origin from science and do not have strong experiences in software engineering and parallel programming (many model teams even do not have any full-time software engineer), we have to make DAFCC as convenient as possible, especially for the model developers who are not proficient in parallel programming and parallel debugging with MPI. So, we try to make DAFCC handle as much work as possible. Now, the MPI communicator of whole ensemble of a component model for running an ensemble DA algorithm is generated automatically and then used intra DAFCC and the data exchanges among members, ensemble and DA algorithm are also automatically handled by DAFCC, no matter the differences regarding parallel decompositions. Moreover, we enable a DA algorithm to be enclosed in dynamic-linking library, in order to make the model code and the DA code as independent as possible.

2. There were 2 main points I could find in section 2, namely "PDAF [... imposes] a precondition of process layout such that each ensemble member uses the same number of processes with successive IDs in the MPI_COMM_WORLD" and "[PDAF] only makes the processor cores of the first ensemble member available to the DA algorithm and forces the processor cores used by other ensemble members to idle when running the DA algorithm". The second statement I think is untrue, but Dr Lars Nerger has posted a short comment on PDAF so I trust he will ensure the correctness there. The first point I find to be obscure as I cannot think of a situation where you would not have ensemble members using sequential MPI process IDs.

Response: The second point is untrue, according to the discussions with Dr Nerger. In PDAF, a DA method can use different processor cores from the first ensemble member, while users should be responsible for developing the data exchange functionalities among different parallel decompositions. Regarding the first point, we also have not seen a real case that ensemble members of a coupled model do not use sequential MPI process IDs. We will not highlight the support for non-sequential MPI process IDs again when revising the manuscript.

3. One overarching question which is not addressed is why would you design from the outset a "weakly coupled" data assimilation system? Why not design a strongly coupled system and then simplify it? I suppose the answer here is to still be able to piggyback on existing observation processing systems and to allow for different observation frequencies, but this should be clearly set out in the article.

Response: Thanks a lot for pointing out this important consideration. We will clearly state it when revising this manuscript.

4. I would like to see more clarity in relation to the comparisons that you make. There are a number of places where the comparison is against an system that uses I/O and reading/writing files from/to disk rather than MPI communications. In such a case phrases like "accelerating the DA system" should be qualified. There are other relations made where it is unclear what the comparison is with. For example in the abstract you state that the new methodology "enables the DA method to utilize more processor cores in parallel execution" but I cannot see the baseline for such a statement. Moreover would such a statement hold with a different baseline?

Response: We will rephrase "accelerating the DA system" and "enables the DA method to utilize more processor cores in parallel execution" that are incorrect.

5. Line 57. PDAF is indeed *the* standard for ensemble based DA frameworks. Others also exist. For example EMPIRE (https://pbrowne.bitbucket.io/empire) Browne, P. A., & Wilson, S. (2015). A simple method for integrating a complex model into an ensemble data assimilation system using MPI. Environmental Modelling \& Software, 68, 122–128. https://doi.org/10.1016/j.envsoft.2015.02.003. You need to discuss other parallel

strategies such as that used by P.A. Browne, S. Wilson, 2015

Response: Thanks a lot for introducing this pioneer work. We will briefly introduce and discuss it when revising the manuscript.

6. Line 93: "How to compile the code of DA methods with the model". This is not necessary. In particular if you run (using MPI) in MPMD mode then the model and the DA could be compiled independently.

Response: It is true that the model and the DA could be compiled independently when using MPMD mode where the DA has its own executable. Regarding DAFCC, the DA does not have its own executable and shares the processor cores of the model ensemble, which means that MPMD mode is not used. The model and the DA could also be compiled independently under DAFCC, because the technique of dynamic linking is used. How to compile DA may be not a critical problem. We will not highlight it again when revising the manuscript.

7. Line 108: "Although PDAF enables a DA algorithm to run in parallel, it only makes the processor cores of the first ensemble member available to the DA algorithm and forces the processor cores used by other ensemble members to idle when running the DA algorithm." This is not my understanding of PDAF. I see that Dr Lars Nerger has already submitted comments in relation to PDAF, so I am assured that he will have given you the latest and correct information in relation to this.

Response: We will revise the manuscript according to the discussions with Dr Nerger.

8. Are you suggesting a coupled model which uses a different coupler, such as OASIS, would then be put into C-Coupler2.0 for the DA component?

Response: We want to offer an option of solution here. C-Coupler2 can automatically generate MPI communicator of each ensemble member of a coupled model that uses a different coupler. There can be other solutions for this functionality, while C-Coupler2 should also be called for recording the MPI communicator of each ensemble member

for using DAFCC.

9. Line 155/Figure 3: Is a restriction that the components in each ensemble members run on the same number of MPI processes? Surely there is a restriction enforced by the DA algorithms that the component model is on the same grid for every ensemble member, or has some very exotic DA methodology been implemented? In the case they have, how do you then establish which DA algorithms are applicable given the difference in the ensemble members?

Response: We agree that there is a restriction enforced by the DA algorithms that the component model is on the same grid for every ensemble member, while a model on the same grid can be run under different numbers of cores as well as different parallel decompositions. Now we note that real cases generally use the identical core number as well as identical parallel decompositions to run ensemble members of the same model. Although we produced the support for inconsistency of core number among ensemble members, it is useless for current real cases. We will not highlight this support again when revising the manuscript.

10. Line 159/160: "execution of a DA algorithm in a component model does not force the processes of other component models to be idled". This must relate to the time stepping procedure of the coupled model. In fact here are you for the fi̧rst time enforcing that all components of the model must have separate MPI processes? This is not the case in, for example, the ECMWF earth system model (Mogensen, K., Keeley, S. and Towers, P., 2012. Coupling of the NEMO and IFS models in a single executable. Reading, United Kingdom: ECMWF.)

Response: We correct the corresponding statement.

11. Line 188: The "weakly coupled" component of your methodology then relies on using the C-Coupler2.0 to control the coupling of the model then?

Response: C-Coupler2 and DAFCC can only handle the coupling between the model

ensemble and the DA algorithm, while the coupling among component models in each ensemble member can also be handle by the original coupler that can be not C-Coupler2.

12. Figure 5: Why is there no red within the DA_CCPL_RUN subroutine to indicate data exchange between the model and the DA?

Response: The data from the model ensemble to the DA algorithm is transferred automatically and implicitly by DAFCC before running the DA_CCPL_RUN subroutine, while the data from the DA algorithm to the model ensemble is transferred automatically and implicitly after running the DA_CCPL_RUN subroutine. This implementation is motivated from some programming languages such as Fortran. We will briefly introduce that when revising the manuscript.

13. Line 410) "We evaluate the effectiveness of DAFCC1 in developing a weakly coupled ensemble DA system". I don't see the justification for this statement. I can see you have implemented the system and shown how it performs computationally with various parameters, as well as a very simplistic verification that the data assimilation is implemented correctly. You should state a measure for effectiveness - was it simply to have a functioning system? Compare this with Browne and Wilson, 2015, where they "propose a simple implementation strategy which does not focus on maximum efficiency of the code. Instead the focus is on the speed of implementation."

Response: In this manuscript, we tried several aspects to evaluate the effectiveness of DAFCC1. First, we adapted an existing ensemble DA system, WRF GSI/EnKF, to DAFCC, where the simulation result of the DA system keeps exactly unchanged. Second, we evaluated the impact of DAFCC in terms of replacing the corresponding I/O operations in the original DA system by MPI. Third, we showed that DAFCC can serve the construction of a weakly coupled ensemble DA system. We will make clear how we evaluate the effectiveness when revising the manuscript. Moreover, considering DAFCC enables a DA algorithm to flexibly utilize a wide range of processor core number (even from 1 to the total core number of the corresponding model ensemble), we will further evaluate the corresponding impact when revising the manuscript.

14. Line 424) Why were 3200 cores used when each node has 24 processors? mod(3200,24) != 0

Response: Although our account can use a maximum number of 3600 cores, we can only use about 3200 cores actually (there are may be some errors in the computer system).

15. Line 460/Figure 11c) Why does the offline timing of GSI vary with different numbers of ensemble members? On line 458 you state that you run all ensemble members of the offline system concurrently, so I would expect a constant value of time for the model run as you change the number of ensemble members. This clarification will be essential in understanding the rest of the figures here, as otherwise it seems like the comparison may be unfair. Could it be i/o related? With every member trying to write output files at the same time your system slows? If this is the case it should be explicitly accounted for in the final paragraph of this section. Furthermore, you should detail what file system architecture is used at BSCC in section 6.1. Is it something like lustre?

Response: We will further discuss about that when revising the manuscript.

Best regards,

Li Liu
* * *

---

## Author Comment (AC1) · 4 Aug 2020

We thank the anonymous reviewers and Dr. Lars Nerger for carefully reading the manuscript and for providing the very valuable comments. The review comments reveal weak points of this manuscript and give us a lot of suggestions for further revision. Guided by the review comments, we will try to significantly improve the manuscript in the following main aspects:

1. Based on the comments of anonymous reviewers and the discussions with Dr. Lars Nerger, we will revise the statements about PDAF in our manuscript and significantly rewrite about our motivation part for developing DAFCC, as the discussions in

the replies to the reviewers.

2. We will add more introductions and references to the related works, such as EMPIRE and the examples of operational coupled data assimilation mentioned by anonymous reviewers.

3. We will clarify or reformulate some statements, such as why the weakly coupled ensemble data assimilation system is designed as the first step target, the specific data assimilation method used in our experiments and "accelerating the DA system" will be rephrased as "enables the DA method to utilize more processor cores in parallel execution", etc.

4. We will make clear how we evaluate the effectiveness, and considering DAFCC enables a DA algorithm to flexibly utilize a wide range of processor core number (even from 1 to the total core number of the corresponding model ensemble), we will further evaluate the corresponding impact when revising the manuscript.

Best regards,

Chao Sun,

on behalf of all authors.

---

## Author Response (AR1)

**Part 1: Responses to Anonymous Referee #1**

1. First, please increase the dpi for the figures, it's hard to read them.

Response: We have increased the dpi for the figures in the revised manuscript.

2. Line 21: please properly introduce the acronyms: "...system GSI/EnKF and the coupled model FIO-AOW".

Response: The acronyms corresponding to GSI/EnKF and FIO-AOW have been added. Please refer to L54 and L311.

3. Lines 26-31: Here you are talking about two widely used DA algorithms: ensemble techniques derived from the Kalman filter and variational data assimilation. But there are methods that you don't mention. Please reformulate the paragraph accordingly. Besides, I would suggest to remove the sentence ", can be viewed as a special case of ensemble-based methods with only one member in the ensemble when we attempt to design and develop a software framework for data assimilation." (lines 30-32).

Response: Some DA methods such as Optimal Interpolation (OI) and Ensemble OI has been added. Please refer to L31~L36. Currently, we do not remove the sentence ", can be viewed as a special case …", and add a new discussion at L471~L474. We will further modify the manuscript if this modification is improper.

4. Line 34: Please add the Environment and Climate Change Canada's (ECCC) hybrid 4D-EnVar DA used operationally in the weather prediction system: Buehner, M., McTaggartᵛ AˇRCowan, R., Beaulne, A., Charette, C., Garand, L., Heilliette, S., Lapalme, E., Laroche, S., Macpherson, S.R., Morneau, J. and Zadra, A. (2015) Implementation of deterministic weather forecasting systems based on ensembleˇ AˇRvariational data assimilation at Environment Canada. Part I: the global system. Monthly Weather Review, 143, 2532–2559

Response: We have modified manuscript accordingly. Please refer to L38.

5.  Lines 36-37: With the rapid development of science and technology, numerical forecasting systems with only an individual component model (such as an atmospheric model) have reached a predictability limit. Please reformulate it or add appropriate references.

Response: We have reformulated this statement with new references. Please refer to L40~L42.

6.  Lines 37-38: Coupled models have been widely used in numerical forecasting to break the bottleneck of the limited predictability... I would also reformulate it in a way that it was established that coupled models may provide better results with respect to uncoupled models. Besides, you may wish to add references for coupled models that used operationally in several operational centres (Met Office, ECMWF, ECCC, etc.)

Response: We have reformulated this statement with new references. Please refer to L40~L42.

7.  Lines 42-43: add references to coupled data assimilation (CDA) systems developed in JAMSTEC, NOAA, Met Office, ECMWF, ECCC: Sugiura et al., 2008, Mochizuki et al., 2016, Yang et al., 2013, Zhang et al., 2014, Lea et al., 2015, Laloyaux et al. 2016 and 2018, Browne et al., 2019, Skachko et al. 2019

Response: The above references have been added into the revised manuscript. Please refer to L45~L48.

8.  Lines 45-49: It is not clear what do you mean. Reformulate or simply remove this sentence.

Response This sentence has been reformulated. Please refer to L49~L52.

9.  Line 49: You don't mention the NASA GMAO system, for example. I would suggest changing "most" by "several".

Response: The manuscript has been modified accordingly. Please refer to L52.

10. Lines 49-60: The ensemble size of current global operational systems used in NWP generally exceeds 256 members. The global models used in the ensemble are run on grids with horizontal sampling of several tens of kilometers. So, the I/O operations are currently necessary. If the aim of the paper is not to discuss real systems, please state it more clearly in the abstract and the beginning of introduction. It will help to understand what kind of CDA system you are developing and its future use.

Response: Thanks a lot for your suggestions. Our statements may not be clear enough to cause misunderstandings. DAFCC1 can handle the original I/O operations for data exchange between model ensemble and DA algorithms via MPI, while the I/O operations of operational systems for outputting results are still kept. Regarding the evaluation in this manuscript, the corresponding I/O operations of WRF as well as other component models are still kept. In the revised manuscript, we try to use the word "extra" for distinction. Please refer to L59~L61.

11. Line 62-63: reformulate the meaning of WCDA. Not the models are assimilated independently, but data from model components are assimilated independently by two separate DA systems.

Response: We have modified the corresponding statement. Please refer to L70~L71.

12. Line 76-78: It's obviously more efficient. However, you don't mention the feasibility of this approach when a model (or several models and couplers) is called in the PDAF Fortran code is a subroutine.

Response: We have modified the corresponding statements. Please refer to L87~L89.

13. Line 332: again, GSI/EnKF has not been defined yet. What is GSI?

Response: We have a brief introduction in Sec 1 L54: "… Grid Point Statistical Interpolation (GSI; Shao et al., 2016) combined with EnKF (Liu et al., 2018a), …". In Sec 5.1, more details about the system structure, operation, integration and driver by DAFCC1 of GSI/EnKF are introduced.

14. Lines 330-343: It's difficult to understand this section. First, you have an atmosphere-ocean-wave coupled model. Second, you have a DA system that you call GSI/EnKF. Does this system compute atmospheric analyses? How ocean analyses are computed? Do you compute also analysis for the waves?

Response: It has been stated and discussed that the example DA system only computes atmospheric analyses corresponding to WRF currently. Please refer to L319~L320, L478~L480.

15. Line345: What do you mean by "combines"? Is it a hybrid system? Or you may choose between two options: a variational (which one?) and ensemble technique? You may also state explicitly that both options share parts of codes, for example, observation operators.

Response: GSI/EnKF is composed of a variational DA sub-system (GSI) and an ensemble DA sub-system (EnKF), which can be used as variational, pure ensemble or hybrid DA systems sharing the observation operator in GSI codes. In this paper, we focus on the pure ensemble DA system for developing the sample DA system, where GSI is only used as the observation operator that calculates the difference between model variables and observations on the observation space and EnKF calculates analysis increments and updates model variables. It provides two options for calculating analysis increments for ensemble DA; i.e., a serial Ensemble Square Root Filter (EnSRF) algorithm and a Local Ensemble Kalman Filter (LETKF) algorithm, while EnSRF is used in this paper. The manuscript has been modified accordingly. Please refer to L324~L326.

16. Lines 345-350: What DA method is used in your experiments: EnKF, EnSRF or ETKF? Please clarify. Instead of using "pure ensemble DA", you may simply say that the EnKF (or ETKF or whatever) mode was chosen.

Response: EnSRF is used in our experiments. This has been stated in the revised manuscript. Please refer to L329~L330.

17. Line 354-371: atmospheric DA? So, do you use EnKF to perform atmospheric analyses?

Response: Yes, we use EnKF to perform atmospheric analyses both in the ensemble DA sub-system of WRF and the example DA system of FIO-AOW. This has been stated in the revised manuscript. Please refer to L330.

18. L373: what step you are talking about? An atmospheric model time step?

Response: "Step" means the flowchart for running the pure ensemble DA system in a DA window shown in Figure 7a and further summarized in Sec4.1.1. We have revised the statements to make it clearer. Please refer to L331~L333 and L354.

19. l.374-395: As shown... It's not clear.

Response: We have revised the related statements accordingly. Please refer to L356~L357.

20. L397: where it was used? And who has upgraded the coupled model to the c-coupler 2.0? After I read the section 5, I realized that there is no ocean data assimilation in your system. So I wonder why you call your prototype WCDA? Your system is an atmospheric DA using a coupled model to compute background states.

Response: FIO-AOW has been used for some typhoon simulations and researches in the referenced works (Zhao et al., 2017 and Wang et al., 2018; please refer to L311~L312). It has been upgraded to C-Coupler2 by us (please refer to L380). It is true that our system is an atmospheric DA using a coupled model to compute background states. We call it WCDA following the WMO report (Penny et al., 2017): "for WCDA, the direct impacts of observations on the analysis are limited to the domain in which the observations reside, while cross-domain impacts are produced as a secondary effect by the integration of the coupled model forecast", which does not restrict whether one-component or multiple-component data are assimilated.

**Part 2: Responses to Anonymous Referee #2**

1. The motivation for developing your own system rather than using PDAF I found lacking. There were 2 main points I could find in section 2, namely "PDAF [... imposes] a precondition of process layout such that each ensemble member uses the same number of processes with successive IDs in the MPI_COMM_WORLD" and "[PDAF] only makes the processor cores of the first ensemble member available to the DA algorithm and forces the processor cores used by other ensemble members to idle when running the DA algorithm". The second statement I think is untrue, but Dr Lars Nerger has posted a short comment on PDAF so I trust he will ensure the correctness there. The first point I find to be obscure as I cannot think of a situation where you would not have ensemble members using sequential MPI process IDs. If you have a specific situation where this is the case, please elaborate on it so the reader can understand why this is important. One overarching question which is not addressed is why would you design from the outset a "weakly coupled" data assimilation system? Why not design a strongly coupled system and then simplify it? I suppose the answer here is to still be able to piggyback on existing observation processing systems and to allow for different observation frequencies, but this should be clearly set out in the article.

Response: Based on the discussions with Dr Lars Nerger, we rewrite the motivation for developing DAFCC. Please refer to L80~L106 in Section 2. A statement has been added for the question why we would design from the outset a "weakly coupled" DA system. Please refer to L69~L70.

2. I would like to see more clarity in relation to the comparisons that you make. There are a number of places where the comparison is against an system that uses I/O and reading/writing files from/to disk rather than MPI communications. In such a case phrases like "accelerating the DA system" should be qualified. There are other relations made where it is unclear what the comparison is with. For example in the abstract you state that the new methodology "enables the DA method to utilize more processor cores in parallel execution" but I cannot see the baseline for such a statement. Moreover would such a statement hold with a different baseline?

Response: The manuscript has been modified accordingly. Please refer to L20~L22, L25, L437, L438 and L471.

3. The article gives a reasonable overview and references for general data assimilation concepts. However the article should point the reader to some of the latest examples of operational weakly coupled data assimilation. Good references for this include, with the first using PDAF:

Goodliff, M., Bruening, T., Schwichtenberg, F., Li, X., Lindenthal, A., Lorkowski, I., & Nerger, L. (2019). Temperature assimilation into a coastal ocean-biogeochemical model: assessment of weakly and strongly coupled data assimilation. Ocean Dynamics, 69(10), 1217–1237. https://doi.org/10.1007/s10236-019-01299-7

Skachko, S., Buehner, M., Laroche, S., Lapalme, E., Smith, G., Roy, F., . . . Garand, L. (2019). Weakly coupled atmosphere-ocean data assimilation in the Canadian global prediction system (v1). Geoscientific Model Development, 12(12), 5097–5112. https://doi.org/10.5194/gmd-12-5097-2019

Browne, P. A., de Rosnay, P., Zuo, H., Bennett, A., & Dawson, A. (2019). Weakly Coupled Ocean-Atmosphere Data Assimilation in the ECMWF NWP System. Remote Sensing, 11(234), 1–24. https://doi.org/10.3390/rs11030234

Response: We have added the recommended references. Please refer to L45~L48.

Specific comments:

4. Lines 12,25. "better" than what?

Response: We have modified the corresponding statement. Please refer to L13.

5. Line 47: "how to conveniently (1) achieve an ensemble run of a coupled model" What is your measure of convenience here? This is a task which is regularly done at many centres around the world, do they all have inconvenient methods for running ensembles of coupled models?

Response: The corresponding statement has been modified. Please refer to L50.

6. Line 55: On the use of disk files, this is also a robust strategy when it comes to massively parallel computing, as this risk of random task failures increases with the size of the coupled models and the number of ensemble members. This should be noted as positive reason for using disk files, as well as the potential to use a larger ensemble than can be run at a single time on an HPC machine.

Response: We make a corresponding discussion in the revised manuscript. Please refer to L496~L506.

7. Line 57. PDAF is indeed *the* standard for ensemble based DA frameworks. Others also exist. For example EMPIRE (https://pbrowne.bitbucket.io/empire) Browne, P. A., & Wilson, S. (2015). A simple method for integrating a complex model into an ensemble data assimilation system using MPI. Environmental Modelling \& Software, 68, 122– 128. https://doi.org/10.1016/j.envsoft.2015.02.003

Response: Thanks a lot for introducing this pioneer work. Brief introductions and discussions about EMPIRE have been added. Please refer to L62~L65 and L80~L87.

8. Line 93: "How to compile the code of DA methods with the model". This is not necessary. In particular if you run (using MPI) in MPMD mode then the model and the DA could be compiled independently.

Response: As how to compile DA may be not a critical problem, we have reduced the related statements in the revised manuscript.

9. Line 108: "Although PDAF enables a DA algorithm to run in parallel, it only makes the processor cores of the first ensemble member available to the DA algorithm and forces the processor cores used by other ensemble members to idle when running the DA algorithm." This is not my understanding of PDAF. I see that Dr Lars Nerger has already submitted comments in relation to PDAF, so I am assured that he will have given you the latest and correct information in relation to this. You need to discuss other parallel strategies such as that used by P.A. Browne, S. Wilson, 2015.

Response: Based on the discussions with Dr Lars Nerger, we rewrite the motivation for developing DAFCC, where EMPIRE has also been briefly discussed. Please refer to L80~L106 in Section 2.

10. Line 115: what are such preconditions? Can you give examples where these exist, and if they do, why they are a problem?

Response: We have removed the statements about preconditions.

11. Line 122: You should make clear this is because the MPI processes from all ensemble members are available. Or are there even more available?

Response: We have modified the corresponding statement, please refer to L103~L104.

12. Line 131: Are you suggesting a coupled model which uses a different coupler, such as OASIS, would then be put into C-Coupler2.0 for the DA component?

Response: Yes, C-coupler2.0 can handle this case.

13. Figure 1: This has no explanation. I fail to see the usefulness of this figure.

Response: For explanations of Figure 1, please refer to L117~L131.

14. Line 143: What is the alternative to DLL?

Response: DLL is the unique choice. We have revised the manuscript accordingly. Please refer to L123.

15. Line 146: "The ensemble component manager is responsible for generating and managing the communicator of ensemble members of a component model." Does this mean you have a separate ensemble component manager for every component of your coupled model, such as atmosphere, ice, land, composition, etc? If so please state this to help the reader.

Response: The ensemble component manager is a common software module shared by all components. We modified "a component model" into "each component model". Please refer to L125~L127.

16. Line 155/Figure 3: Is a restriction that the components in each ensemble members run on the same number of MPI processes? Surely there is a restriction enforced by the DA algorithms that the component model is on the same grid for every ensemble member, or has some very exotic DA methodology been implemented? In the case they have, how do you then establish which DA algorithms are applicable given the difference in the ensemble members?

Response: We have deleted this part in the revised manuscript.

17. Line 159/160: "execution of a DA algorithm in a component model does not force the processes of other component models to be idled". This must relate to the time stepping procedure of the coupled model. In fact here are you for the first time enforcing that all components of the model must have separate MPI processes? This is not the case in, for example, the ECMWF earth system model (Mogensen, K., Keeley, S. and Towers, P., 2012. Coupling of the NEMO and IFS models in a single executable. Reading, United Kingdom: ECMWF.)

Response: We have revised the corresponding statement to make it clearer. Please refer to L134~L135.

18. Line 165). Point 4. Please give an examples of such a DA algorithm procedures. Do you mean, for example, that DA algorithm 1 procedure 1 would be calculation of model equivalents ($H(x)$) and DA algorithm 1 procedure 2 would be something like an SVD of the ensemble perturbation matrix?

Response: Yes. For example, for the GSI/EnKF system, GSI can be used as DA algorithm 1 procedure 1, which is used as the observation operator that calculates the difference between model variables and observations on the observation space; EnKF can be used as the DA algorithm 1 procedure 2, which calculates analysis increments and updates model variables of all ensemble members. Please refer to the example shown in L359~L362.

19. Line 169: "Scripts are allowed to conduct necessary process control". This comes out of the blue, and it is not clear how this fits within the methodology of having everything using MPI communication. Can you give examples in section 4.4.

Response: This statement is incorrect. We have revised it. Please refer to L144~L145.

20. Line 188: The "weakly coupled" component of your methodology then relies on using the C-Coupler2.0 to control the coupling of the model then?

Response: It does not rely on using the C-Coupler2.0 for model coupling. C-Coupler2 and DAFCC can only handle the coupling between the model ensemble and the DA algorithm, while the coupling among component models in each ensemble member can also be handle by the original coupler that can be not C-Coupler2.

21. Figure 5: Why is there no red within the DA_CCPL_RUN subroutine to indicate data exchange between the model and the DA?

Response: We have added more introductions to the DA_CCPL_RUN subroutine. Please refer to L192~L195.

22. Section 4.4, 2). Terms such as periodic timer, period_unit, period_count and lag are introduced with no context. These should be defined as well as an explanation of why they are needed for data assimilation.

Response: The periodic timer is used to enables users to flexibly set the frequency as well as the model time of running the corresponding DA algorithm (L290~L291). The introduction to periodic timer can be found in L291~L295.

23. Section 4.4) I fail to see why any of this is relevant to data assimilation. What is an example of statistical processing in a DA context?

Response: Two kinds of statistical processing of input model fields are supported in DAFCC1: the statistical processing in each time window named "time_processing" and the statistical processing among ensemble members named "ensemble_operation". "Time_processing" specifies whether the field values from the model to the DA algorithm are instantaneous or time averaging. Time averaging model values may be required in some interdecadal DA experiment. "Ensemble_operation" refers to whether the fields transferred from the model to the DA algorithm is the value of a single member or the values of all ensemble members. For example, in the flowchart for running GSI/EnKF (Sec 4.1.1), the model is required to provide the mean values of all ensemble members to GSI in the third main step and the aggregated values of all ensemble members to EnKF in the firth main step. The manuscript is modified accordingly. Please refer to L296~L299.

24. Line 330) "A sample weakly" - do you mean "An example weakly coupled ensemble DA system"? I don't know what "sample" refers to here. This is used many times throughout the manuscript - please clarify.

Response: "sample" has been replaced by "example" throughout the paper.

25. Section 6.1) The details of the EnSRF (i.e. localization radius and inflation factors) are not useful without a description of the model.

Response: Descriptions of the model parameters are given in the following part of Sec 5.1 (for example, horizontal resolution can be referred to Table 1). Some details of the EnSRF in this part are introduced for the reproducibility of the experiment, although the DA experiment in this paper is mainly used to verify DAFCC1 and we do not pay too much attention to the evaluation of actual DA effect.

26. Line 410) "We evaluate the effectiveness of DAFCC1 in developing a weakly coupled ensemble DA system". I don't see the justification for this statement. I can see you have implemented the system and shown how it performs computationally with various parameters, as well as a very simplistic verification that the data assimilation is implemented correctly. You should state a measure for effectiveness - was it simply to have a functioning system? Compare this with Browne and Wilson, 2015, where they "propose a simple implementation strategy which does not focus on maximum efficiency of the code. Instead the focus is on the speed of implementation."

Response: In fact, we focus on the code speed and correctness of DAFCC1 in developing a weakly coupled ensemble DA system. "effectiveness" has been replaced by "correctness" throughout the paper.

27. Line 424) Why were 3200 cores used when each node has 24 processors? mod(3200,24) != 0.

Response: Although our account can use a maximum number of 3600 cores, we can only use about 3200 cores actually (there are may be some errors in the computer system).

28. Line 436) "variables used for DA" -> "are the prognostic, or analysed, variables in the data assimilation".

Response: Thank you for your suggestions. We have revised the statement, please refer to L416~L418.

29. Section 6.2 could be a simple statement saying that WRF-GSI/EnKF with DAFCC1 is bit-identical to the original offline WRF-GSI/EnKF.

Response: We have added introduction to the validation standard. Please refer to L433~L434. We still keep most original content in Section 5.2 because we think that we should show why we employ the bit-identical standard for validation.

30. Line 460/Figure 11c) Why does the offline timing of GSI vary with different numbers of ensemble members? On line 458 you state that you run all ensemble members of the offline system concurrently, so I would expect a constant value of time for the model run as you change the number of ensemble members. This clarification will be essential in understanding the rest of the figures here, as otherwise it seems like the comparison may be unfair. Could it be i/o related? With every member trying to write output files at the same time your system slows? If this is the case it should be explicitly accounted for in the final paragraph of this section. Furthermore, you should detail what file system architecture is used at BSCC in section 6.1. Is it something like lustre?

Response: We add a discussion about the above issues in Section 5.3 (L456~L460). The Lustre file system is used at BSCC (L402~L403).

31. Section 6.4) This is not a measure of the effectiveness in developing a weakly coupled ensemble DA system. Figure 15 shows results for northern hemisphere (25N-90N) and tropics (25S-25N) but I understood this was a limited area system, running from 0N-50N. You should update the figure to reflect this. All this figure appears to show is that the DA system is producing increments which have the correct sign.

Response: "Effectiveness" has been replaced by "correctness" throughout the paper. There were mistakes in the title in Figure 15, which has been corrected (L781~L782).

Part 3: a marked-up manuscript version

[revised manuscript text omitted]

---

## Author Response (AR2)

First, we'd like to say many thanks to the editor and the reviewers. The review comments give us a lot of suggestions for further improving the manuscript. The manuscript has been revised accordingly. In the following, we will reply the review comments one by one.

Best regards,

Chao Sun, on behalf of all authors.

**Part 1: Responses to Topical Editor**

1. I agree with referee #2 that the clarity and structure of the manuscript still needs to be improved. This is especially the case for sections 2 and 3. A suggestion from my side would be to first provide a general introduction how the C-Coupler works in a technical sense (e.g. with an example for two models) mentioning required coding steps for coupling, details on data transfer etc. Then you could introduce the changes you made for allowing ensemble propagation and the use of DA algorithms and explain the details of the implementation. Some additional comments and suggestions are given below. I would like to ask you to include these comments in a further revision of your manuscript.

Response: Thanks a lot for your suggestions. A new figure (Fig. 1 in the revised manuscript) about the model flowchart with C-Coupler2.0 and new model flowchart with ensemble DA based on C-Coupler2.0 is added to Section 2, which introduces the major model steps for achieving coupling exchanges among component models with C-Coupler2.0 and the new steps for achieving coupling exchanges between a DA algorithm and a model ensemble. We have modified the corresponding statements. Please refer to P4L114~L122.

2. Line 35-36: This statement is a bit misleading as it gives the impression that from a methodological point of view variational methods are a subset of ensemble methods, which is not quite correct. To me it seems that the statement is more from a technical perspective (--> managing the ensemble propagation). I would suggest to rephrase this statement in order to avoid confusion.

Response: We have rephrased the statement. Please refer to P2L35~L36.

3. Line 40: involving --> evolving

Response: We have corrected it.

4. Line 52-53: What do you mean with 'coupled DA system' in this context? Does 'coupling' refer here to DA with coupled models or to the coupling of a DA framework to a model? For example, DART is a quite generic DA framework, that can do DA with any kind of model (be it a single compartment model or a coupled earth system model). Please specify.

Response: Here we refer to the ensemble DA frameworks supporting coupled DA. We have modified the statement. Please refer to P2L52~L53.

5. Line 93-96: I would not call this 'extra coding'. The choice of processor/ communicator layout is a regular part of the implementation of a DA system (e.g. the coupling of PDAF with a user code) and this somehow depends on the users needs and preferences. It's just that the template files in PDAF include a standard example with the described processor layout that can be relatively easily adapted to user needs as suggested by Dr. Nerger's short comments.

Response: We have revised the statements of this section. Please refer to P3L88~L95.

6. Line 96: COMM_COUPLE, COMM_FILTER, etc. is a quite specific nomenclature for PDAF which is probably not known to the reader. Please explain this in a more general sense.

Response: We have added general descriptions about the specific communicators for PDAF. Please refer to P3L93~L95.

7. Line 101-102: '..which does not use global communications,...': It is not quite clear to me what you mean here (i.e. what 'global communication' you refer to). Ensemble filters always require a step where the data from the different realizations are collected. How would the implementation with C-Coupler differ in that respect (as compared to other already existing frameworks)?

Response: Global communications here refers to communications among all processes in the global communicator, e.g., MPI_gatherv and MPI_scatterv. Although ensemble filters always require a step for collecting the data from the different realizations of a model, a process of the ensemble filter only calculates the analyses on a part of grid points, and thus a process only requires to collect the data from a part of processes of the model. PDAF and DAFCC1 adopt this communication mode. We have revised the corresponding statements. Please refer to P4L101 and P4L108.

8. Line 102-103: '...and does not require users to develop extra codes...': With reference to your statements in lines 93-96, how does the implementation with C-Coupler differ in that respect? Please provide more details how your implementation can handle different processor layouts for models and DA algorithms. For example, in Figure 2, how would C-Coupler handle a case where you want to run DA algorithm 1 with all available processors instead of the ones shown in the figure?

Response: More details about the data exchanges between models and the DA algorithms are introduced in P8L236~L246, which can be easily achieved by the import/export interfaces of C-Coupler2.0, as they are all coupled as component models.

9. Line 92/103: It is not clear what you mean with 'process sets'. Please provide more details.

Response: "process set" means a set of processes. The manuscript has been modified accordingly. Please refer to P3L93 and P4L101.

10. Line 123: Please explain in more detail what you mean with the DLL technique.

Response: More details about the DLL technique have been added into the manuscript. Please refer to P4L128~P5133.

11. Line 168: What do you mean with 'ensemble-set component model'? Please explain more clearly.

Response: We have added more introductions to 'ensemble-set component model'. Please refer to P6L174~L176.

12. Line 201-207: Does that mean that the DA software component is restarted at every assimilation cycle (as suggested by keywords initialize, run, finalize)?

Response: The DA software component can be reused across assimilation cycles with restart. The manuscript has been modified accordingly. Please refer to P7L213~L216.

13. Section 5.4: This is rather short. Please provide a more detailed analysis and description here as a main focus of the paper (already mentioned in the title) is on weakly coupled DA systems.

Response: We have added more analysis of the results from the weakly coupled DA system (including Fig. 18). Please refer to P16L482-L485.

14. Line 468-470: '...while still guaranteeing software independence between model and the DA method.': As I understand the coupling approach, you would still need to adapt the model code when using a different DA software (as suggested e.g. by lines 364-377). Please clarify.

Response: We have revised this statement. Please refer to P16L489~L490.

**Part 2: Responses to Anonymous Referee #2**

1. The introduction contains now a reasonable overview and references of the currently used coupled data assimilation systems. The motivation of the authors seems also clearer than is the previous version. Though, you could add some words about your intention to provide a flexible tool for the scientific community that needs minimum programming efforts (as you did in the response to the referee). However, my major concern of the previous version persists. Except the introduction, there is no difference with the previous version. The text of the article is not structured, especially the section 3, contains long difficult-to-understand sentences. Many acronyms and terms are not properly introduced. The authors do not provide a reasonable description of the systems and models used, but mention some data assimilation parameters like localization length scale and inflation factors that doesn't make sense in isolation. I would suggest you to put some additional efforts on the text of the article.

Response: Thanks a lot for your suggestions. Combined with Topical Editor's comments and suggestions, we have added a new figure (Fig. 1 in the manuscript) about the model flowchart with C-Coupler2.0 and new model flowchart with ensemble DA based on C-Coupler2.0 in section 2, which introduces the major model steps for achieving coupling exchanges among component models with C-Coupler2.0 and the new steps for achieving coupling exchanges between a DA algorithm and a model ensemble. Please refer to P4L114~L122 for more details. We have checked and added more introductions about some acronyms or special terms in Section 3, e.g., P6L175~L176. The details of some DA parameters are mentioned for the reproducibility of the experiment, and we have also added more introductions about the coupled model. Please refer to P11L329~L330.

2. Besides, introduce properly your weakly coupled data assimilation. What is the data assimilation system number one, say atmospheric, what is number two (ocean or wave DA?). How the DA systems talk to each other? Did you put the calls for both DA systems into the PDAF code? Maybe, a good idea would be to add a figure showing the scheme of the WCDA. Please work also on the description of the experiments, extend the discussion of results and conclusions.

Response: We have added a new figure (Fig. 10 in the manuscript) about the architecture of FIO-AOW and the corresponding weakly coupled ensemble DA system, where the atmospheric analyses are computed by GSI/EnKF that has been coupled with the ensemble of WRF based on DAFCC1, while each ensemble member of other component models is impacted by the atmospheric analyses via model coupling. Please refer to P13L395~L398 for more details.

3. L39 : … on an ensemble run

Response: We have revised it to "…the ensemble run", please refer to P2L39 in the manuscript.

4. L100 : To develop a new framework… The fact that you mention here a WCDA may be misleading for readers. Flexible manipulation of MPI tasks may be beneficial for any sequential data assimilation.

Response: Our motivation for this work is from WCDA, while DAFCC1 can also be beneficial for the DA of a single-component-model system (Section 4.1 and Section 5.3 is an example).

Part 3: a marked-up manuscript version

[revised manuscript text omitted]

**(a)** FIO-AOW MPI Processes

| 0 | 1 | 2 | 3 | 4 | 5 | 6 |

| 0 | 1 | 2 | 3 | 4 | 5 | 6 |

| 0 | 1 | 2 | 0 | 1 | 0 | 1 |

| MPI_COMM_WORLD |
| FIO-AOW |
| WRF |
| POM |
| MASNUM |

**(b)** DA instances

| GSI observer for member |
| GSI observer for ensemble mean |
| EnKF |

**(c)** MPI Processes

[revised manuscript text omitted]

---

## Author Response (AR3)

**Part 1: Responses to Topical Editor**

Thank you for submitting a revised version of your manuscript. According to reviewer #1's and my own assessment, the concerns regarding clarity of the manuscript from the former review round were only partly addressed by your revision. Please have a close look at suggestions and examples in the referee report of reviewer #1. A few additional suggestions for improvements are also given below.

Please take some time to go thoroughly again through the whole manuscript and try to improve wording and explanations so that the text is easier to understand for the reader.

Response: Thanks a lot for handling our manuscript and for your comments. Combined with the comments from former review round and the referee report of reviewer #1, we further go thoroughly again through and revised our manuscript to make it clearer and easier to understand. We'd like to reply the comments one by one.

Specific comments:

1.    Line 31: Sentence incomplete.

Response: We have rephrased the sentence. Please refer to P1L31~P2L36.

2.    Line 42: "...Earth system models...".

Response: We have corrected it. Please refer to P2L42.

3.    Line 75: "...is described...".

Response: We have corrected it. Please refer to P3L75.

4.    Line 83: "...global communication with ...".

Response: We have revised the related statements. Please refer to P3L82 and P4L100.

5.    Line 94: "communicators for the model, the filter and the coupling exchanges between them...". It is probably still not clear to the reader what you refer to here. Please explain in more detail.

Response: We have removed the description of PDAF communicators which is not easy to understand for readers. We then rephrased the statements. Please refer to P3L91~L94.

6.    Line 98: "...waiting for the results...".

Response: We have corrected it. Please refer to P4L97.

7.    Line 99-102: Sentence unclear. Please rephrase.

Response: We have rephrased the statements. Please refer to P4L98~L104.

8. Line 117: "The most significant..." or "A significant...".

Response: We have corrected it. Please refer to P4L118.

9. Line 118: "...or intra one...". Wrong wording. Use something like "...or within one...".

Response: We have revised the related statements. Please refer to P4L119 and P8L238.

**Part 2: Responses to Anonymous Referee #1**

My concerns about the general structure, especially the sections 2 and 3, of the paper have not been addressed. The text is still extremely difficult to understand. Below are my specific comments to the text of the section 2 and 3 to demonstrate the frustration of a reader. Besides, the WCDA implementation has not been properly introduced despite my previous concerns. I don't think that this article may be published in its current state in the GMD journal.

Response: Thanks a lot for your concerns and suggestions. In this revision, we have tried to improve the manuscript according to your comments.

Specific comments:

1. l.84: Maybe like this? Such an implementation maintains the independence between the DA and model modules. But the global communications are generally inefficient in the sequential DA systems because of idle processes due to sequential running of model and DA modules.

Response: We have rephrased the statements. Please refer to P3L83~L85.

2. L86: I don't understand these two sentences: In PDAF, a DA method is transformed into a native procedure that is called by the corresponding models via the PDAF application programing interfaces (APIs). Thus, a DA method can share the processes of the model ensemble.

Response: In EMPIRE, a DA method cannot share the processes of the model ensemble, because the DA method is compiled into a standalone executable and two different executables cannot share processes in MPI run. In PDAF, a DA method is transformed into a native procedure of a model, which also means

that the DA method and the model are compiled into the same executable. Thus, a DA method can share the processes of the model ensemble. Please refer to P3L85~L88.

3. L89: DA method shares all processes of the first ensemble member of the corresponding model… Not clear.

Response: We have rephrased the statements with more introductions. Please refer to P3L87~L91.

4. L101. …enables a DA method to share almost all the processes of the model ensemble… not clear

Response: We have rephrased the statements with more introductions. Please refer to P4L100~L102.

5. L102. When a DA algorithm uses processes different from a model ensemble member… not clear

Response: We have revised the statements. Please refer to P4L105~L107.

6. l.106. I would modify this sentence like this: Fortunately, such a challenge has already been overcome by most existing couplers (Craig et al., 2012; Valcke, 2012; Liu et al., 2014; Craig et al., 2017; Liu et al., 2018b). Each of these couplers (?) can transfer data between different process sets with different parallel decompositions without the global communications.

Response: Thanks a lot for your careful suggestions. We have rephrased the sentence. Please refer to P4L108~L110.

7. L116-118. Something shorter like: Combining multiple components into one MPI program is challenging because CC2 can handle exchanges between two components only.

Response: Thanks a lot for your suggestions. C-Coupler2.0 can handle coupling exchanges between two component models or within one component model no matter it is one MPI program or not. But C-Coupler2.0 cannot directly handle coupling exchanges between a DA algorithm and each model ensemble member as their process sets are partially overlapping. We have rephrased the sentence. Please refer to P4L118~L120.

8. L120. Please explain how these three new steps help to solve the mentioned problem.

Response: We have added more introductions about our solutions to the challenge. Please refer to P4L120~L126.

9. L121. The sentence 'These three steps enable all members in a model ensemble to use a DA algorithm cooperatively.' is not clear. Please reformulate it.

Response: We have rephrased the sentence. Please refer to P4L125~L126.

10. l.127 A DA algorithm can include a set of procedures such as observation operators and analysis modules, each of which can be called by the model separately. I would only say: Each module of the DA algorithm may be called separately in the PDAF system. Is this what you are saying here?

Response: We have rephrased the sentence. Please refer to P5L131~L132.

11. L128-130. Instead of 'The framework uses the dynamic link library (DLL) technique for the connection of a DA algorithm program to a model, where a DA algorithm program is compiled into a DLL that is dynamically linked to a model when an instance of the DA algorithm is initialized.' I would say: 'The dynamic link library (DLL) technique is used to connect a DA algorithm program to a model program. The DA algorithm program is compiled into a DLL that is dynamically linked to a model program.'

Response: Thanks a lot for your careful suggestions. We have rephrased the statements. Please refer to P5L132~L134.

12. L130-133. Instead of 'With the DLL technique, a new DA algorithm can be used by a model without modifying and recompiling the model codes, and the original configuration and compilation systems of a DA algorithm can generally be preserved for greater independence of the DA algorithms from the models and for less work in integrating a DA algorithm.' I would say 'Using the DLL technique allows us to couple DA algorithm and model codes without modifying the codes.'

Response: Thanks a lot for your careful suggestions. We have rephrased the statements. Please refer to P5L134~L136.

13. L133-134. The ensemble component manager governs the communicators of ensemble members.

Response: We have rephrased the sentence. Please refer to P5L136~L137.

14. L134-135. The online DA procedure manager provides several APIs that enable the ensemble members of a component model to initialize, run and finalize a DA instance cooperatively. Not clear. And this part of the sentence may be a separate sentence 'automatically handles the data exchanges between the ensemble members and the DA algorithm'. You can delete 'with a set of operations'.

Response: We have rephrased the statements. Please refer to P5L137~L140.

15. L136-138. This is not clear: The ensemble DA configuration manager enables the user to flexibly specify the DA algorithm, DA frequency and the operations for the data exchange in a DA

simulation through a configuration file. What is DA frequency? What is 'to specify the DA algorithm', what are 'the operations for the data exchange'?

Response: We have rephrased the statements. Please refer to P5L140~L141.

16. L139. Not clear: Guided by the architecture in Fig. 2, we implemented the new framework

Response: We have rephrased the statements. Please refer to P5L142~L143.

17. L143. DA algorithms at different frequencies… What do you mean?

Response: Here, different frequencies refer to different time periods to calling DA algorithm program. We have rephrased the statements. Please refer to P5L145~L147.

18. L143. while component 3 does not use DA. Not clear.

Response: We have rephrased the statements. Please refer to P5L147.

19. L158-161. Not clear.

Response: We have removed some difficult-to-understand terms about C-Coupler2.0 and rephrased the related statements. Please refer to P6L161~L163.

20. L169-170. Given an ensemble run of a coupled model, all ensemble members of the component models of the coupled model can be organized as one level of models. Not clear.

Response: We have rephrased the related statements. Please refer to P6L171~L173.

21. L170-174 although we recommend constructing two hierarchical levels of models with 170 the first level corresponding to all ensemble members of the coupled model and each ensemble member including the component models at the second level (Fig. 5b), because the hierarchical organization retains the original architecture of the coupled model through a simple additional registration of the coupled model to C-Coupler2.0.. Not clear.

Response: We have rephrased the related statements. Please refer to P6L172~L176.

22. L174-176. As a DA algorithm that handles ensemble fields can run on the MPI processes of all ensemble members of a component model (Fig. 3), a special C-Coupler2.0 component model that covers all ensemble members of the component model (this 175 special component model is called ensemble-set component model hereafter) is required for using the DA algorithm (Fig. 5b). Not clear.

Response: We have rephrased the related statements. Please refer to P6L177~L179. For the definition of ensemble-set component model, please refer to P4L120~L121.

23. L177-178. The ensemble component manager provides the capability to generate an ensemble-set component model, which does not introduce global synchronization and only involves the ensemble members of the corresponding component model. Not clear.

Response: We have rephrased the related statements. Please refer to P6L180~L183.

24. L180. A pair of a model and a DA algorithm have essentially the relationship between a caller and a callee in a program. Why is so?

Response: We have rephrased the related statements. Please refer to P6L185~L186.

25. L183 what is native code?

Response: We have removed the statement. Please refer to P6L188~L189.

26. L183 a corresponding compiler… not clear

Response: We have rephrased the related statements. Please refer to P6L188~L189.

27. L185. What is host model?

Response: A model that calls a DA algorithm is called the host model of the DA algorithm. Please refer to P6L187~L188.

28. L186. To address the above challenge... The challenge is not clear to me.

Response: Here, the challenge is that when a DA algorithm is enclosed in a DLL and dynamically linked to the host model, compilers cannot guarantee the consistency of the argument list between the host model and the DA algorithm. We have rephrased the related statements. Please refer to P6L188~L191.

29. I stopped reading here. This article would have been an excellent reference for the future users of such complicated system. However, I don't see how it can be helpful for them due to such unclear text.

Response: Besides the revisions of above specific comments, we further go thoroughly again through our manuscript, especially the sections 2 and 3, and we have revised most long difficult-to-understand sentences and added more introductions about some acronyms or special terms to make it clearer and easier to understand. Please refer to sections 2 and 3 for more details.